# Unveiling chromatin dynamics with virtual epigenome

Ming-Yu Lin[1], Yu-Cheng Lo[1] & Jui-Hung Hung [1,2] ✉

The three-dimensional organization of chromatin is essential for gene regulation and cellular function, with epigenome playing a key role. Hi-C methods have expanded our understanding of chromatin interactions, but their high cost and complexity limit their use. Existing models for predicting chromatin interactions rely on limited ChIP-seq inputs, reducing their accuracy and generalizability. In this work, we present a computational approach, EpiVerse, which leverages imputed epigenetic signals and advanced deep learning techniques. EpiVerse significantly improves the accuracy of cross-cell-type Hi-C prediction, while also enhancing model interpretability by incorporating chromatin state prediction within a multitask learning framework. Moreover, EpiVerse predicts Hi-C contact maps across an array of 39 human tissues, which provides a comprehensive view of the complex relationship between chromatin structure and gene regulation. Furthermore, EpiVerse facilitates unprecedented in silico perturbation experiments at the "epigenome-level" to unveil the chromatin architecture under specific conditions. EpiVerse is available on GitHub: https://github.com/jhhung/EpiVerse.

The spatial organization of chromatin within the nucleus—encompassing how DNA is packed and interacts within its three-dimensional environment—dictates the accessibility of genetic information to the cellular machinery, thereby influencing gene expression patterns[1]. These patterns are pivotal for orchestrating cellular processes, guiding development, and modulating responses to environmental cues. Furthermore, the dysregulation of chromatin architecture is increasingly recognized as a contributing factor to disorders, including cancer, highlighting the need for a detailed understanding of its mechanisms[2-4]. Techniques such as high-throughput chromosome conformation capture (Hi-C) have revolutionized our ability to map the three-dimensional architecture of the genome[5-8]. These methods have unveiled the fundamental organization of chromosome territories and identified functional domains such as topologically associating domains (TADs) and chromatin loops[9,10].

Despite the insights provided by Hi-C and related methodologies, they are not without significant limitations. The cost and labor intensity of these techniques, combined with their requirement for extensive genomic material, restrict their application to large-scale or high-throughput studies[11]. This limitation is particularly acute when considering the exploration of chromatin dynamics under various epigenetic conditions, where the need for repeated measurements under different experimental conditions or across diverse cell types escalates the required resources significantly.

Predictive modeling of chromatin conformation represents a promising avenue to circumvent some of these challenges as it aims to computationally simulate DNA folding and its regulatory effects. However, existing models, which rely solely on sequences or a limited range of chromatin immunoprecipitation sequencing (ChIP-seq) inputs for chromatin interaction inference in restricted scenarios, often fail to capture the underlying characteristics responsive to chromatin dynamics[12-15]. This results in limited accuracy and reduced generalizability across different cellular contexts. Attempts to improve these models by incorporating more ChIP-seq data or other epigenetic markers face obstacles due to the scarcity of the necessary ChIP-seq data. This scarcity further constrains their capacity to accurately model chromatin behavior across a range of conditions, perturbations, or over time.

[1]Department of Computer Science, National Yang Ming Chiao Tung University, HsinChu, Taiwan, ROC. [2]Program in Biomedical Artificial Intelligence, National Tsing Hua University, HsinChu, Taiwan, ROC. ✉e-mail: jhh@cs.nycu.edu.tw; juihunghung@gmail.com

We introduce the concept of the "virtual epigenome" in a novel computational framework, called EpiVerse, for understanding chromatin dynamics. Unlike traditional epigenetic approaches, the virtual epigenome encompasses computationally imputed epigenetic data constructed through advanced machine learning techniques that integrate diverse datasets, including but not limited to, histone modification patterns, transcription factor (TF)-binding profiles, RNA expression, and DNA accessibility data, and can be constructed with as few as one ChIP-seq track. By leveraging the power of imputation, we synthesize comprehensive epigenetic profiles from sparse or incomplete datasets, thus creating a "virtual" representation that retains the complexity and informational richness of actual epigenetic states.

In this work, we demonstrate that EpiVerse outperforms existing models in five performance metrics, indicating its superior ability to preserve the structural integrity of chromatin architecture. Furthermore, by embedding the chromatin state prediction task within a multi-task learning model, EpiVerse substantially enhances the model's interpretability, facilitating a deeper and more nuanced comprehension of chromatin dynamics. Through its ability to predict chromatin contact maps across an extensive variety of human tissues, EpiVerse affords a comprehensive insight into the complex interplay between chromatin structure and gene regulation, uncovering consistent patterns that transcend tissue differences. This not only expands our understanding of chromosomal organization but also elucidates fundamental principles of gene expression and regulation, such as the impact of chromatin folding on gene accessibility, as observed across multiple tissue types. Additionally, EpiVerse facilitates in silico perturbation experiments, allowing researchers to simulate the effects of "genome-level" (e.g., clustered regularly interspaced short palindromic repeats [CRISPR editing) and "epigenome-level" (e.g., ChIP-seq profiles indicative of metastatic cell state transformations) perturbations on chromatin structure. This capability offers unprecedented opportunities to explore chromatin dynamics under specific conditions, advancing our understanding of gene regulation and chromatin architecture across diverse cellular contexts.

## Results

### An overview of EpiVerse's framework

In brief, the EpiVerse pipeline unfolds through a structured three-phase process, commencing with the Avocado model[16]. Avocado is adept at uncovering and interpreting latent representations within the human epigenome. A select set of epigenetic profiles is input into Avocado to extrapolate a comprehensive epigenomic landscape, which encompasses 71 distinct epigenetic signals. These virtual epigenetic signals, when integrated with one-hot encoded DNA sequence data, are forwarded to the subsequent stages of the EpiVerse pipeline, as depicted in Fig. 1a.

The second phase introduces HiConformer, an innovative component that combines a diagonal extraction algorithm with multi-task learning to enhance contact map prediction accuracy and interpretation. The diagonal extraction algorithm enables the model to deduce chromatin contacts between two regions by analyzing information from all intervening regions, which has been found beneficial in predicting chromatin contacts[13]. By extracting diagonals from the Hi-C matrix and utilizing a sliding window technique, HiConformer efficiently processes features up to a 0.5 Mb region, capturing both individual and interaction-pair characteristics within the chromatin architecture (Fig. 1b).

Furthermore, HiConformer's multi-task learning framework allows for the simultaneous imputation of ChromHMM states and Hi-C contact maps. This methodology not only aids in identifying features that contribute to functionally relevant annotations but also ensures the model's predictive accuracy for contact maps. The integrated approach adopted by HiConformer significantly enriches our understanding of the epigenome's role in shaping chromatin structure.

The operation of HiConformer begins with an encoding phase where epigenetic and DNA sequence data are transformed into embeddings via a multi-layer Convolutional Neural Network (CNN). These extracted embeddings are then concatenated and serve as the input for two distinct task branches within the model. For the ChromHMM prediction task, the genomic embedding is directed through fully connected layers designed to predict chromatin states. Concurrently, for the Hi-C prediction task, the genomic embedding is fed into a BERT Transformer block[17], specialized for generating predictions of Hi-C contact probabilities (Fig. 1c). These predictions are subsequently refined through a downstream denoising module, ensuring high fidelity in the reconstructed chromatin contact maps.

In the final stage, we adopt MIRNet[18] to denoise the diagonals generated by the HiConformer (Fig. 1d), which contains upstream imputation artifacts. We first assemble these diagonals into 1 M matrices and then leverage MIRNet's capabilities in real image restoration and enhancement to refine the Hi-C data, enhancing the detail and clarity of Hi-C matrices. This denoising step is crucial in reducing artifacts from both the data and the imputations from the upstream pipeline process. The resultant Hi-C contact map, combined with the genome sequence, virtual epigenome, and chromatin states, provides a holistic depiction of chromatin that we term the "virtual epigenetic landscape".

### Evaluation of EpiVerse across cell types

To train and evaluate the EpiVerse pipeline, we began by curating high-quality datasets, focusing on the IMR90, GM12878, and K562 cell lines −well-established benchmarks in Hi-C model training and evaluation. Leveraging the ENCODE Consortium's existing imputations by Avocado, we retrieved comprehensive epigenetic datasets, including 71 high-quality tracks[16,19,20] (Supplementary Data 1). These tracks covered a broad spectrum of epigenetic markers, including histone modifications, TF-binding sites, RNA expression profiles, and DNase-seq. For the subsequent stages of the EpiVerse training pipeline, we collected the 3D-genome Interaction Viewer and database (3DIV) distance-stratified normalization Hi-C datasets[21] (Supplementary Data 2). We also incorporated the ROADMAP Epigenomics ChromHMM 25-state dataset to enhance the multi-task learning with chromatin state prediction tasks, complementing the Hi-C predictions by offering insights into chromatin states across various cell types[22].

For each cell line, chromosomes 1 through 18 were designated for the training set, chromosomes 19 and 20 for the validation set, and chromosomes 21 and 22 were used as the testing set, ensuring a consistent and segregated approach to model training, validation, and testing across all cell lines. When using the Avocado model, we employed its pre-trained weights directly for cell lines already included in the Avocado dataset. For cell lines not covered, we fine-tuned the Avocado model to reflect their unique epigenetic conditions. The HiConformer model underwent training to minimize a hybrid loss function, tailored to align its predictions with the actual Hi-C and ChromHMM data from the target cell lines. This hybrid loss function addresses the inherent distance bias present in Hi-C datasets and enhances HiConformer's accuracy in predicting various ChromHMM states, resulting in a comprehensive and precise representation of chromatin structure and states. Lastly, the MIRNet module, which processes diagonals predicted by HiConformer, was trained using a novel surrogate loss function. This loss combines diagonal correlation, which captures distance-stratified Hi-C relationships, and Charbonnier loss for robustness against outliers. The surrogate loss uniquely balances prediction accuracy with structural consistency, improving the quality of interaction data.

Next, we assessed EpiVerse's performance and robustness across different cell types. Using IMR90 as the training target and the whole genome of GM12878 for testing, we benchmarked EpiVerse against several state-of-the-art methods[12–14]. This evaluation included

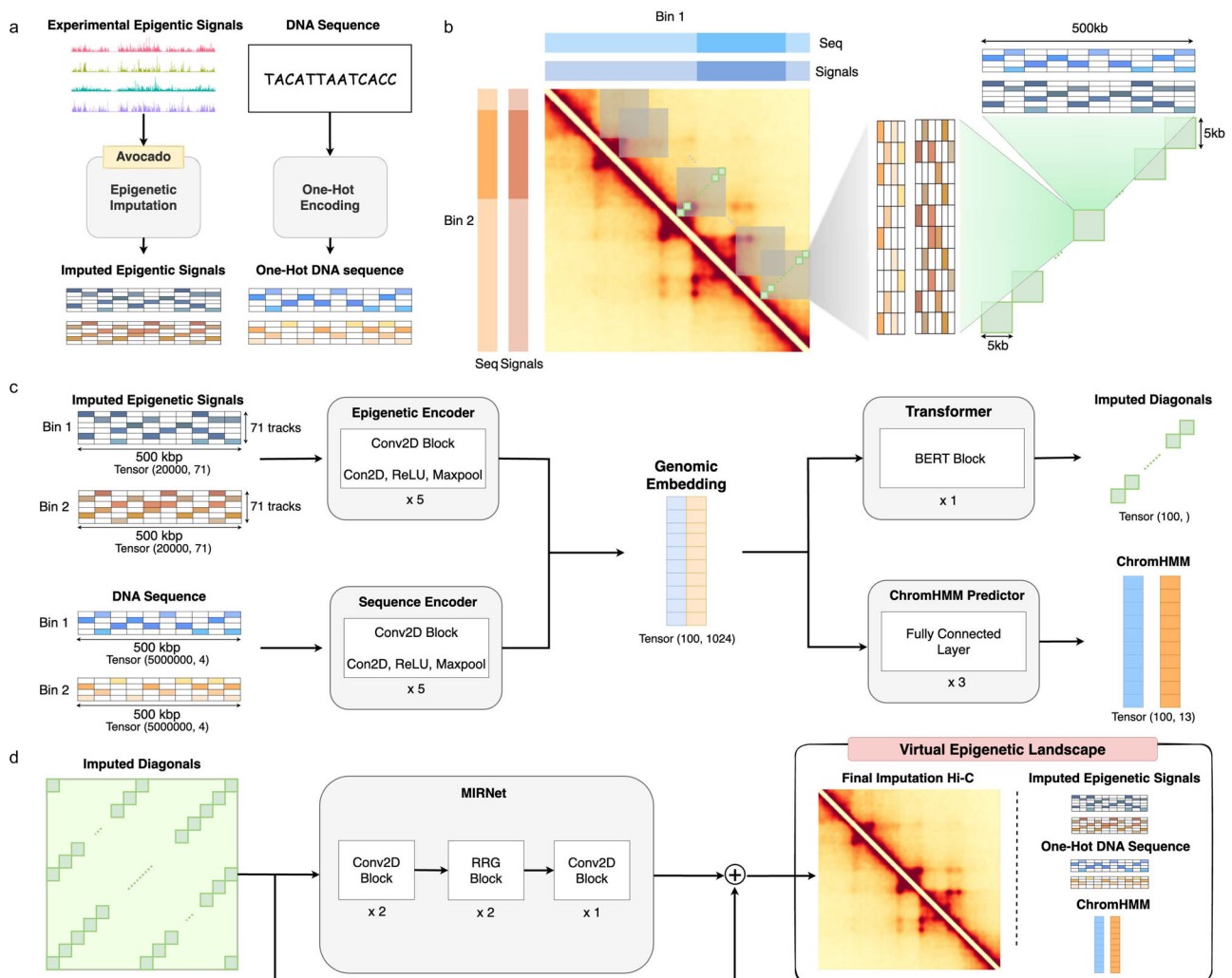

**Fig. 1 | Overview of the EpiVerse pipeline components and workflow.**
**a** Epigenetic signal and DNA sequence generation: representation of the imputation process using the Avocado model, which imputes a wide array of epigenetic signals, and their integration with a one-hot encoded DNA sequence to provide detailed epigenome information. **b** Diagonal extraction algorithm: A specialized approach used by HiConformer to enhance the receptive field of features for improved prediction of long-range interactions in Hi-C data, capturing both individual and contextual interaction features across a 0.5 Mb region. **c** HiConformer model: The multi-task CNN-Transformer framework within EpiVerse, which processes and integrates epigenetic signals and DNA sequence data for the imputation of ChromHMM states and Hi-C contact maps, optimizing the prediction of Hi-C and providing biological annotations. **d** MIRNet framework: The denoising component of EpiVerse that applies real image restoration techniques to Hi-C data diagonals, enhancing detail and clarity, and reducing biases from the data and imputation models.

comparisons with Orca, a deep learning framework that predicts multiscale genome interactions using only sequence as input[14]. Another method, Hi-C-Reg, offers a robust framework for generating high-resolution contact count profiles, relying exclusively on epigenetic information to capture both individual locus-level interactions and higher-order genomic organizational units[13]. EpiVerse was also compared to C.Origami, a deep neural network model adept at de novo cell-type-specific chromatin architecture predictions. C.Origami combines DNA sequence data with cell type-specific features such as CTCF and Assay for Transposase-Accessible Chromatin using sequencing (ATAC-seq) signals for its predictive tasks[12].

We conducted a percentile rank-based visualization comparative analysis, comparing predictions from all four methods within distinct percentile ranks: PR99, PR75, PR50, and PR25 (Fig. 2a). Our results reveal EpiVerse's capacity to consistently generate high-fidelity imputations, especially within the PR99 regions, where it closely mirrors the ground truth and preserves essential structural features such as loops and TADs. Even in the PR25 regions, EpiVerse consistently outperforms

other methods. Overall, visual inspection clearly shows that EpiVerse delineates TAD boundaries and details with greater accuracy, while significantly reducing the artifacts commonly associated with other methods.

To verify that the incorporation of comprehensive epigenetic signals is crucial for high-quality cross-cell-type prediction, we conducted an ablation study on our model under conditions analogous to those employed by C. Origami and Orca. Specifically, we ran experiments using only DNA sequence data and a combination of CTCF and DNase as inputs. In both scenarios, we observed a notable decrease in performance (Supplementary Fig. 1), with the quality of the contact maps deteriorating across various percentile ranks (Supplementary Fig. 2). This ablation study underscores the critical importance of incorporating a diverse set of epigenetic signals for accurate chromatin structure prediction.

To systematically evaluate the EpiVerse pipeline's performance in cross-cell-type prediction, we applied multiple metrics: Pearson correlation coefficient, Spearman correlation coefficient, distance-stratified

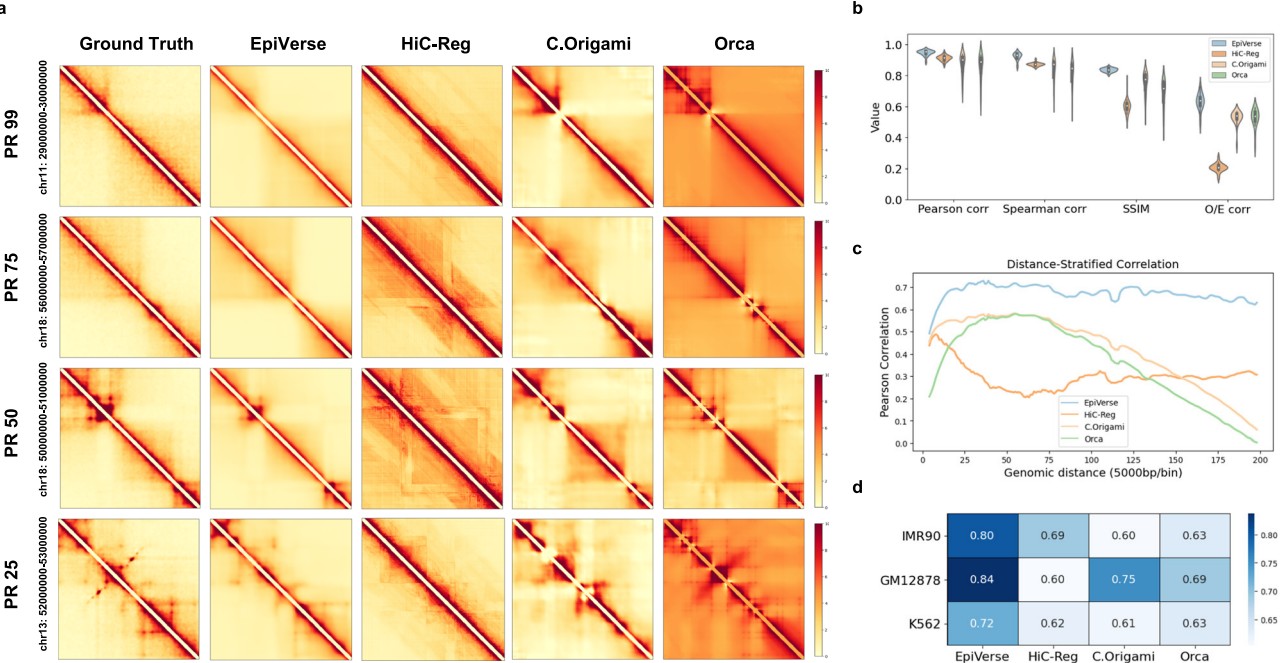

**Fig. 2 | Comparative analysis of EpiVerse with benchmark methods. a** Predicted Hi-C matrices from four methods in four percentile ranges: The visualization compares the ground truth Hi-C interaction matrices with those imputed by Epi-Verse, Hi-C-Reg, C.Origami, and Orca across various percentile ranks (PR99, PR75, PR50, PR25). EpiVerse's predictions are shown to closely approximate the ground truth, particularly in the top 0–25% range, indicating high-quality imputation of chromatin interactions. **b** Performance metrics comparison: The violin plots depict the distribution of Pearson correlation coefficient, Spearman correlation coefficient, SSIM, and O/E correlation values for each method (*n* = 5243), with EpiVerse outperforming other models across all metrics, signifying its superior predictive accuracy. Violin plot elements are defined as follows: center point represents the

median; box limits indicate the upper and lower quartiles; whiskers extend to 1.5 times the interquartile range from the quartiles; and the overall shape represents the kernel density estimation of the data distribution. **c** Distance-stratified correlation comparison: This figure displays Pearson correlation coefficient values stratified by genomic distance, illustrating EpiVerse's enhanced performance in predicting long-range interactions, particularly evident at distances beyond 0.5 Mb. **d** Cross-cell type prediction validation: The table shows the SSIM performance for EpiVerse, Hi-C-Reg, C.Origami, and Orca across different cell types (IMR90, GM12878, K562), with EpiVerse consistently delivering high-quality predictions, demonstrating robust cross-cell-type generalization.

correlation, and structural similarity (i.e., SSIM) across each 1 Mb matrix of the whole-genome region. EpiVerse outperformed other methods in Pearson correlation coefficient, Spearman correlation coefficient, observed/expected (O/E) correlation (Fig. 2b). The distance-stratified correlation, which calculates correlations based on varying genomic distances, further underscored EpiVerse's proficiency, especially notable with a 0.6 correlation at distances beyond 0.5 Mb. This suggests Epi-Verse's capability in capturing long-range interactions, potentially a result of our diagonal extraction algorithm and comprehensive epige-netic signals (Fig. 2c). Notably, EpiVerse outperforms existing models in all performance metrics, indicating its superior ability to maintain chromatin structural integrity. Additionally, we further assessed Epi-Verse's performance by using IMR90-trained models to predict chro-matin structure in the same cell type (IMR90, chromosomes 21 and 22) and in cross-cell types (entire genomes of GM12878 and K562). EpiVerse delivered consistent and high-quality predictions across cell types (Fig. 2d).

The generalization of EpiVerse was demonstrated by varying the training datasets among IMR90, K562, and GM12878 cell lines, Epi-Verse consistently maintained stable performance, affirming its adaptability and robustness across diverse training environments (Supplementary Figs. 3–5). Moreover, EpiVerse exhibited high repro-ducibility; retraining the model multiple times yielded consistently reliable results (Supplementary Fig. 6). The loss curve further con-firmed that EpiVerse converges well during training (Supplementary Fig. 7). Additionally, in evaluating its capability to predict ChromHMM states, EpiVerse achieved an average accuracy rate of over 90% (Sup-plementary Fig. 8), demonstrating its proficiency in utilizing epigenetic signals and capturing the complexities of chromatin state dynamics.

## EpiVerse identifies key chromatin regulatory elements

Identifying elements crucial for chromatin structure and under-standing their role in gene regulation are key objectives in 3D-genome studies. EpiVerse stands out in this endeavor, offering a unique com-bination of the virtual epigenome and chromatin state annotation within the virtual epigenetic landscape. This combination enhances the interpretability of discovering comprehensive elements that define chromatin structure. We showcased IMR90-trained EpiVerse's ability to discern elements that define chromatin structure by focusing on chromosome 21 in K562 and GM12878 cells, which were not exposed to the model during training. The EpiVerse pipeline was applied to four tasks: DNA sequence motif discovery, assessment of the importance of epigenetic signals for chromatin structure using the Integrated Gra-dient (IG)[23], analysis of the impact of epigenetic signals on enhancer states, and exploration of the interplay between enhancers and chro-matin structure.

For DNA sequence motif discovery, we analyzed the first-layer CNN activations of the sequence encoder in HiConformer (Supple-mentary Fig. 9; see Methods). These activations were filtered and processed to construct Position Weight Matrices (PWMs)[24], which represent the nucleotide frequency at each position within a motif, capturing sequence variability across multiple binding sites. The PWMs derived from the CNN filters correspond to the consensus sequences identified by the individual CNN kernels, effectively cap-turing recurring patterns or motifs in the sequence data (Fig. 3a). Interestingly, we observed that most motifs identified by the sequence encoder CNN are not present in the imputed epigenetic signals. This suggests that the features extracted by the sequence encoder are potentially compensating for the features that are not provided by the

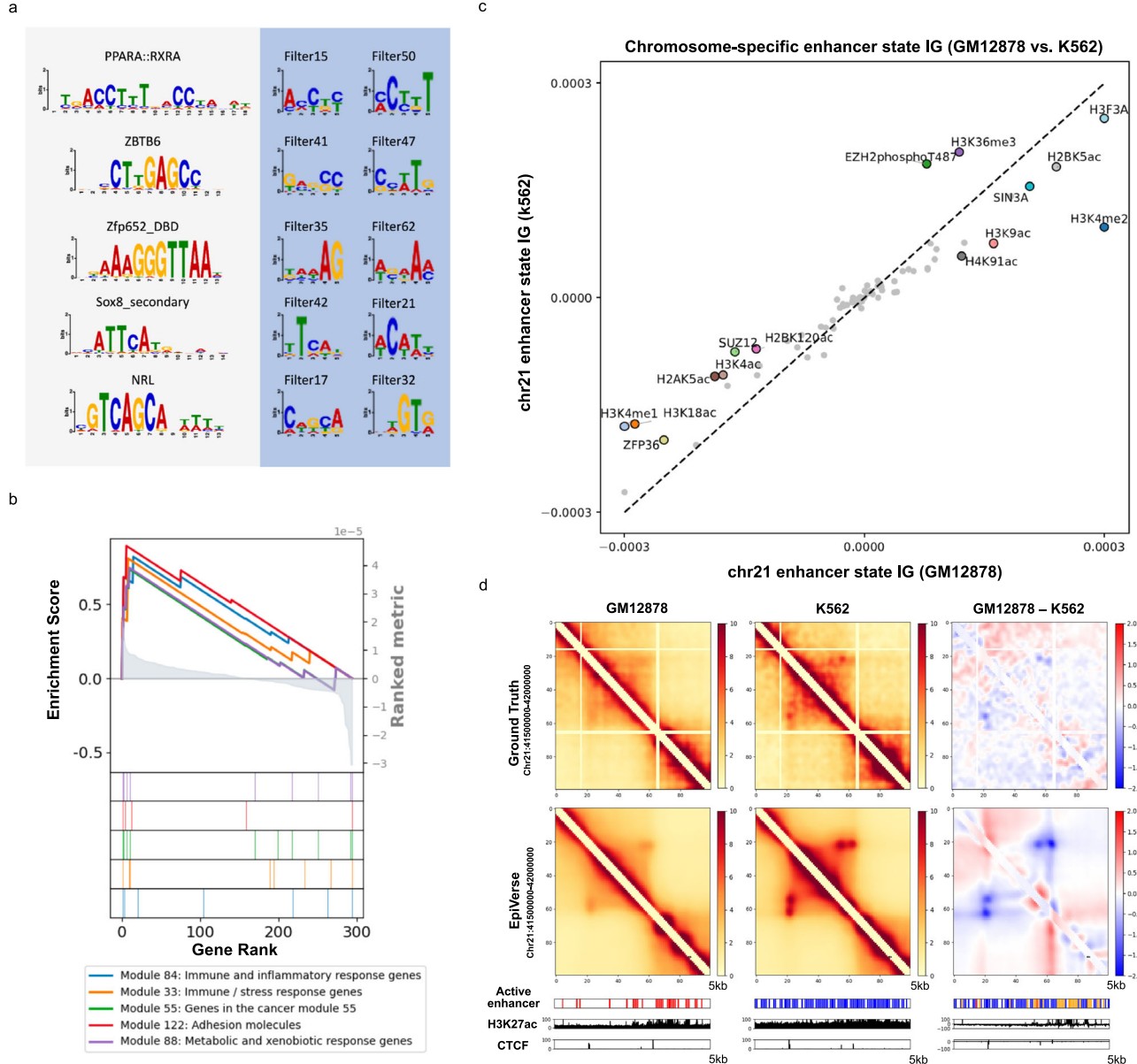

**Fig. 3 | EpiVerse identifies elements defining chromatin structure. a** DNA motif analysis results: The top hit motifs are derived from 64 filters, showcasing the significant DNA sequences identified through the motif analysis. **b** Gene Set Enrichment Analysis (GSEA) Using IG for Chromosome 21: Chart displaying the enrichment scores of genes based on the importance of surrounding epigenetic signals as determined by IG for chromosome 21. **c** Epigenetic signals' tissue specificity in enhancer state: Scatter plot delineating the tissue specificity of various epigenetic signals in their influence on enhancer states, with proximity to the diagonal line indicating the degree of tissue specificity. **d** Largest differential enhancer regions: The heatmaps display the chromatin interaction differences, highlighting regions with significantly varied enhancer activity between the two cell lines. Stronger chromatin looping interactions are evident in K562 as compared to GM12878. The top row represents the ground truth chromatin interactions, while the bottom row shows the imputed data. The rightmost column presents the differential heatmaps, where red indicates regions with stronger interactions in GM12878 and blue denotes stronger interactions in K562. Additionally, the figure includes active enhancer states along with H3K27ac and CTCF signal tracks for the same genomic region. Yellow bars indicate shared enhancer states between the two cell lines.

virtual epigenomes (Supplementary Data 3). This finding implies that the sequence encoder in HiConformer is capturing unique aspects of the genomic data, contributing to a more comprehensive understanding of the genomic features beyond what is revealed by epigenetic signals alone. This complementary relationship between the sequence-derived motifs and the epigenetic signals underscores the complexity of genomic regulation and the importance of integrating multiple data types for a holistic understanding.

For assessment of the importance of epigenetic signals for chromatin structure, we calculated the IG of epigenomic signals from HiConformer (Supplementary Fig. 9; see Methods). IG is a technique used to attribute the contribution of individual features (in this case, epigenetic signals) to a model's prediction[23]. Given that the chromatin structure plays a critical role in cell type-specific transcriptional regulation[25], we attempt to use our epigenetic signal IG scores to identify cell type differential genes. For each gene, we calculated its importance based on the mean IG within a ±0.25 Mb region surrounding the gene promoter. We then compared these gene importance values between K562 and GM12878 cell lines, focusing on the differences to identify differential genes with distinct importance across these cell types. By ranking differential gene importance revealed by IG and conducting a pre-ranked gene set enrichment

analysis (GSEA)[26], we observed that K562 is enriched in cancer-related gene sets[27], consistent with its nature as a leukemia cancer cell line, whereas GM12878 is a normal lymphocyte cell line. GSEA is a computational method used to determine whether predefined sets of genes show statistically significant, coordinated differences between two biological conditions, providing insights into underlying biological processes. Notably, the top five scoring modules according to normalized enrichment scores (NES) are all associated with leukemia (Fig. 3b and Supplementary Data 4). This convergence of epigenetic signal importance and gene set enrichment underscores the critical link between epigenetic landscape and phenotype, further illustrating the power of EpiVerse in uncovering the epigenetic underpinnings of cell identity and pathology.

Building upon our methodology to quantify the impact of epigenetic signals on chromatin structure, we further assessed the genome-wide differential importance of genes, particularly those influencing oncogenesis. This was achieved by calculating the importance values of genes based on the IG method considering the contribution of epigenetic modifications within and around gene promoter regions associated with oncogenic signatures from Molecular Signatures Database (MSigDB)[27]. By meticulously comparing these importance values between cell lines, we identified several oncogenesis-related genes that exhibit a heightened significance in K562 cells, suggesting their pivotal role in the leukemia phenotype of this cell line. (Supplementary Data 5). For example, BMI1 is overexpressed in various types of leukemia, which is related to the signature BMI1_DN_MEL18_DN.V1_DN. BMI1's overexpression is associated with increased cell proliferation and the inhibition of apoptosis and differentiation, which are hallmark traits of cancer cells[28]. Another notable signature was *PDGF_ERK_DN.V1_DN*, *PDGF A* and *PDGF B* are genes induced during the megakaryoblastic differentiation of K562 cells[29]. This differentiation process leads to the development of megakaryocytes, key players in platelet production. The association of these genes aligns with the biological processes observed in K562 cells. Focusing on cancer modules (marked by MSigDB computational gene sets[27]), we found a significant number of modules linked to leukemia (Supplementary Data 6). These results show the efficacy of HiConformer's IG scores in reflecting changes in tissue-specific chromatin structure and their subsequent impact on gene regulation, thus providing crucial insights into the dynamics of 3D-genome organization.

Next, we focused on enhancers, which are known to be tissue-specific, and the activity is largely dependent on chromatin accessibility and TF binding[30–32]. We specifically calculated the IG of epigenetic signals on active enhancer states in K562 and GM12878 cells. Our analysis illustrates the importance of each epigenetic signal on the enhancers of these cell types (Fig. 3c). The distance from the central diagonal line indicates the degree of tissue specificity. We identified SUZ12 as being particularly tissue-specific in K562 cells. SUZ12, a polycomb gene, is known for its overexpression in the bone marrow of patients with chronic myeloid leukemia (CML) in the blastic phase[33]. Additionally, our findings indicate that histone modifications such as H3K36me3 and H3K4ac are enriched in K562 cells, which might be indicative of their cancerous nature[34,35]. Our approach presents a novel method for assessing the impact of epigenetic signals on enhancers delineated by predicted chromatin states, and how these signals influence chromatin structure across different tissues.

To further delve into the dynamic interplay between enhancers and chromatin structure, we undertook a comparative analysis between K562 and GM12878 cells. Our focus was particularly on identifying regions showing the largest disparities in active enhancer activity. In K562 cells, the region that emerged with the most significant increase in active enhancers, relative to GM12878 cells, exhibited notably more pronounced chromatin looping, as illustrated in Fig. 3d. This finding emphasizes that variations in enhancer activity

are not merely incidental but are associated with distinct chromatin configurations across different cell types. It illustrates the dynamic and consequential relationship between enhancer activity and chromatin structure, suggesting that enhancers play a critical role in defining the three-dimensional organization of the genome. EpiVerse's ability to accurately predict and visualize these distinct chromatin configurations and enhancer activities across different cell types demonstrates its values in advancing our understanding of the genome's three-dimensional organization.

## EpiVerse maps tissue-specific chromatin interactions

EpiVerse facilitates the analysis of whole human 3D chromatin structure, leveraging its robust cross-cell-type prediction capability. A Hi-C-based dendrogram constructed using hierarchical clustering of 39 representative tissues provided significant insights into the organization of tissue types, illustrating their similarities based on chromatin interaction patterns (Fig. 4a and Supplementary Data 7). In this dendrogram, distinct clusters indicate functional or developmental relatedness among the grouped tissues. Notably, the cluster related to embryonic stem (ES) cells exhibits higher Hi-C interaction frequencies, potentially signaling regions of open chromatin conducive to the transcriptional flexibility required for pluripotency and lineage specification. This enhanced interaction frequency within the ES-cells-related cluster aligns with the active chromatin state commonly observed in stem cells, supporting a broad range of gene expression programs essential for stem cell maintenance and differentiation[36]. Similarly, other distinct clusters, such as those associated with T cells, the esophagus, and B cells, may represent unique chromatin structural signatures linked to their specialized functions. This analysis reveals that chromatin interaction patterns are intricately organized and reflect the specific functional demands and regulatory environments characteristic of each cell type.

To further investigate the relationship between chromatin structure and enhancer activity across different tissues, we conducted a quantile-based analysis, sorting Hi-C interaction values into four quantiles across multiple tissues. Our results reveal a clear positive correlation between the number of active enhancers and Hi-C interaction frequencies, with a notable increase in interaction frequencies from the lowest to the highest quantile (see Fig. 4b and Supplementary Fig. 10). This trend underscores the link between chromatin structure and enhancer activity. Furthermore, we observed a positive correlation between Hi-C interaction frequencies and gene counts, indicating that regions with fewer Hi-C interactions tend to contain fewer genes (see Supplementary Fig. 11). These observations provide further evidence of the critical impact that chromatin architecture has on gene regulation[25].

To further analyze the impact of gene expression on chromatin 3D structure, we conducted virtual 4C analyses targeting the promoters of tissue-specific genes. Our findings indicate a strong correlation between promoter hubness—a metric quantifying the number of connections a promoter engages in—and gene expression[37]. Specifically, genes highly expressed in the brain exhibit significantly higher promoter hubness in the brain than in other tissues, indicating a denser network of chromosomal interactions within the brain context. This correlation between promoter hubness and gene expression is consistent across various tissues, including the pancreas, liver, stomach, skeletal muscle, and smooth muscle, as shown in Fig. 4c. This underscores the tissue-specific nature of promoter hubness and its profound influence on gene expression levels, highlighting the specificity of chromosomal interaction networks in tissue-specific gene regulation. Furthermore, genes situated within regions of the largest variation in chromatin structure, as identified by Gene Ontology (GO) term analysis, demonstrate tissue specificity (see Fig. 4d and Supplementary Fig. 12), further emphasizing the critical role of chromatin architecture in gene regulation.

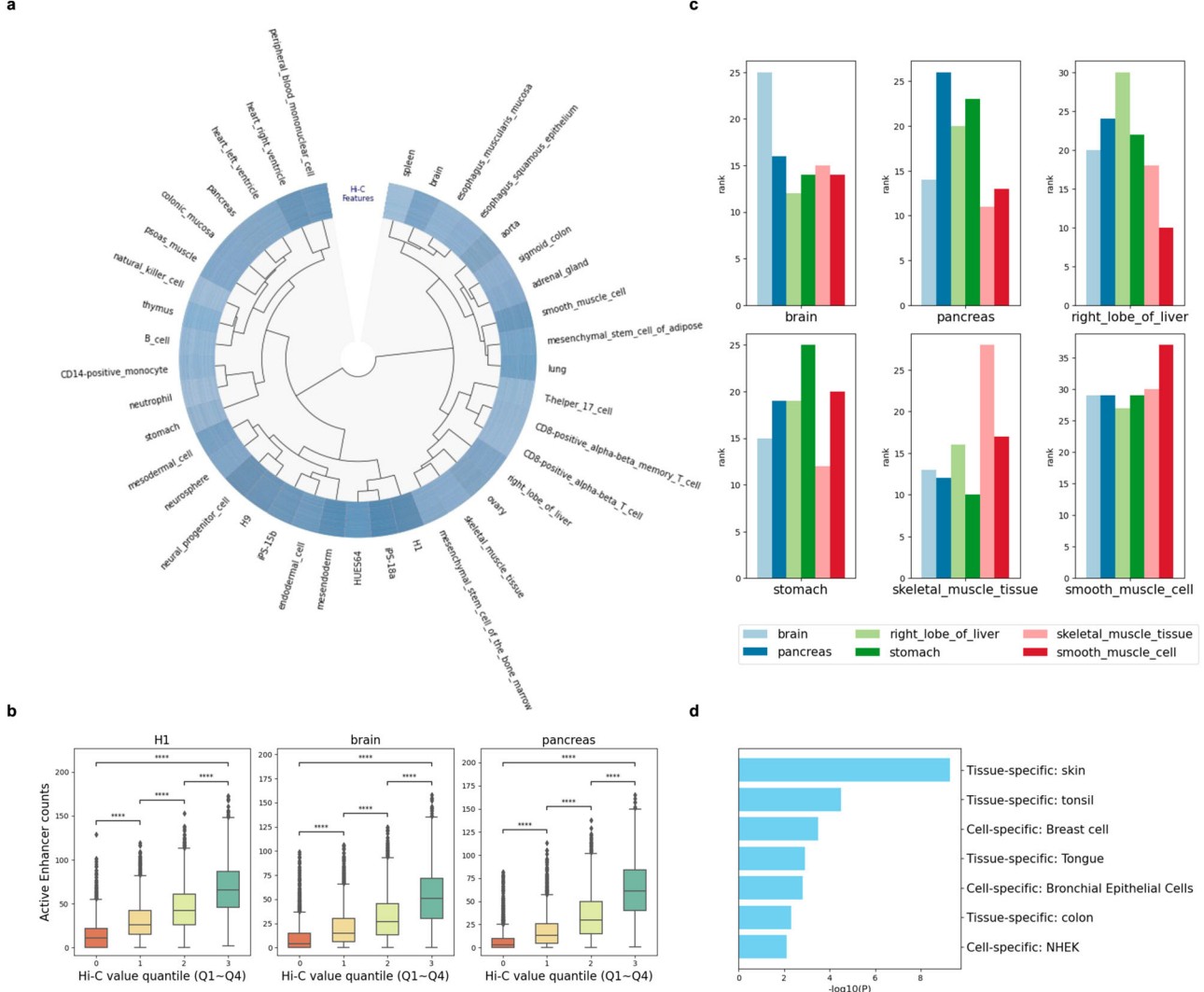

**Fig. 4 | EpiVerse enables the analysis of whole human chromatin structure. a** Hi-C-based dendrogram: A dendrogram illustrating hierarchical clustering of 39 different tissue types based on similarities in their Hi-C features, revealing potential functional or developmental relationships. **b** Boxplot of Hi-C value quantile and active enhancer counts: Boxplot compares the distribution of Hi-C interaction values across four quantiles to the counts of active enhancers ($n = 11,152$; data divided into quartiles), highlighting a trend that suggests a relationship between chromatin interaction frequency and enhancer activity. The two-sided Mann–Whitney test was conducted, with significance indicated by asterisks: *$p < 0.05$, **$p < 0.01$, ***$p < 0.001$, and ****$p < 0.0001$. Pairwise comparisons between groups for three tissues: H1 (0 vs. 1, $p = 2.068 \times 10^{-255}$; 1 vs. 2, $p = 9.135 \times 10^{-112}$; 2 vs. 3, $p = 3.487 \times 10^{-176}$; 0 vs. 3, $p = 0$), brain (0 vs. 1, $p = 5.679 \times 10^{-165}$; 1 vs. 2, $p = 4.01 \times 10^{-89}$; 2 vs. 3, $p = 7.845 \times 10^{-168}$; 0 vs. 3, $p = 0$), pancreas (0 vs. 1, $p = 3.641 \times 10^{-225}$; 1 vs. 2, $p = 4.224 \times 10^{-180}$; 2 vs. 3,

$p = 4.968 \times 10^{-251}$; 0 vs. 3, $p = 0$). Boxplot elements are defined as follows: center line represents the median; box limits indicate the upper and lower quartiles; whiskers extend to 1.5 times the interquartile range from the quartiles; and points denote outliers. **c** Promoter hubness in tissue-specific genes across six tissues: A visual representation of promoter hubness, which reflects the number of connections a promoter has, across six different tissues, showing its correlation with the expression of tissue-specific genes. **d** GO term analysis of genes in regions with the largest Hi-C contact variation: GO term analysis of genes located in regions with the largest variation in chromatin structure highlights strong tissue specificity. This figure demonstrates the critical role of chromatin organization in defining tissue- and cell-specific functions, with significant enrichment in skin, tonsil, breast, tongue, bronchial epithelial cells, colon, and normal human epidermal keratinocytes (NHEK).

To investigate whether both promoter hubness and enhancer activity serve as reliable indicators of tissue-specific gene expression, we selected 19 of the 39 imputed tissues that are also available in the Genotype-Tissue Expression (GTEx) Portal[38]. We then employed weighted Kolmogorov–Smirnov testing using GSEA to evaluate the effectiveness of promoter hubness and active enhancers in identifying tissue-specific gene expression profiles. A tissue was considered a "hit" if its gene ranking by Virtual 4 C promoter hubness or active enhancers appeared in the top five significance results, indicating that the ranked genes successfully identified their own gene expression profile through GSEA. Our analysis found that relying solely on promoter hubness for gene ranking correlated less with tissue-specific expression than expected, achieving only two hits out of 19 tissues.

Conversely, active enhancers proved to be a superior metric, more accurately ranking genes and serving as a more reliable indicator for characterizing tissue-specific gene expression patterns, as demonstrated in Supplementary Fig. 13 (12 hits out of 19 tissues). These findings suggest that active enhancers demonstrate greater tissue specificity across various tissues compared to promoter hubness. This observation is in line with recent studies showing that numerous genes can undergo significant changes in 3D connectivity without corresponding changes in transcription levels[37].

To empower researchers to freely explore and study the relationships between different tissues, we have provided an EpiVerse browser. Accessible at https://epiverse.jhhlab.tw, the EpiVerse browser offers an interactive interface for users to input whole-genome

locations and visualize the corresponding virtual epigenetic landscape (Supplementary Fig. 14). It also supports differential analysis between tissues, simplifying the examination process by enabling direct comparison between two tissues' chromatin structures. This browser is a valuable resource for researchers delving into the intricate dynamics of tissue-specific regulatory mechanisms (Supplementary Fig. 15).

## EpiVerse enables in silico chromatin perturbation

Building on the effectiveness of the virtual epigenetic landscape and the insightful analysis of chromatin architecture's influence on gene regulation and the importance of epigenetic signals empowered by EpiVerse, another prominent innovative capability of EpiVerse is to enable "epigenome-level" in silico perturbation-based Hi-C experiments, offering researchers a unique opportunity to explore the effects of genetic alterations and epigenetic modifications without the need for extensive and resource-intensive laboratory experiments. Demonstrating robust capacity for conducting in silico perturbation experiments at both 'sequence' and 'epigenome' levels, EpiVerse introduces a novel approach to genomic analysis.

At the sequence level, mirroring the success of existing methods like Akita[39] and Orca[14], EpiVerse allows direct manipulation of DNA sequences, such as excision or insertion at specific genomic locations and implication of changes in chromatin structures. For the insertion experiment, we were inspired by a pivotal study that highlighted the importance of engineering chromosomal contact domains[40]. This study successfully demonstrated the creation of contact domains by inserting a 2-kb DNA sequence at a stable domain boundary, which incorporated a CTCF-binding site and a transcription start site. Emulating this approach, we duplicated the epigenetic signals and DNA sequence from this 2-kb region and inserted them into the genomic locations reported from previous experiments. This in silico perturbation-based Hi-C experiments by EpiVerse mirrored the results of the original study, yielding new chromatin contacts and reinforced domain boundaries, as evidenced by the congruence with real Hi-C experiment observations (Supplementary Fig. 16). Additionally, drawing inspiration from a study illustrating the impact of deletions on insulated neighborhood boundaries, which leads to activation of *LMO2* and *TAL1* oncogenes[41], our simulations entailed the removal of both epigenetic signals and DNA sequences at designated sites within these regions. EpiVerse adeptly replicated the results of laboratory deletion experiments, notably within the LMO2-insulated neighborhood (Supplementary Fig. 17). Together, these findings highlight EpiVerse's capacity to model genomic alterations with high fidelity.

At the "epigenome-level" in silico perturbation-based Hi-C experiments, we used EpiVerse to replicate the chromatin architecture alterations observed between primary pancreatic cancer cells (PANC-1) and metastatic pancreatic cancer cells (Capan-1)[42]. These changes, intimately linked to epigenetic state perturbations, affect the formation and reorganization of contact domains and chromatin loops. By utilizing a limited dataset of six epigenetic signals (CTCF, H3K27me3, H3K4me3, H3K9me3, H3K27ac, H3K36me3) from two types of pancreatic cancer cells, PANC-1 (primary) and Capan-1 (metastatic), EpiVerse inferred missing epigenetic tracks using Avocado, integrating these with experimental signals. With HiConformer and MIRNet, it then imputed the chromatin structure, reconstructing an extensive 71-track epigenetic signal array for accurate chromatin structure predictions (Fig. 5a).

EpiVerse's efficacy was further validated by its accurate modeling of alterations in a region located in chromosome 15, recapitulating chromatin loops connecting enhancers to the *LIPC* gene in Capan-1 cells—a feature absent in PANC-1 cells and associated with *LIPC*'s upregulation in Capan-1. Despite losing some fine-scale features, EpiVerse successfully captured the principal alterations within this region (Fig. 5b). Further validation across chromosome 15 revealed that while one track is sufficient for Hi-C contact map prediction (as per the

original Avocado paper), incorporating more experimental tracks improves accuracy (Fig. 5c).

To investigate the relationship between chromatin interaction changes and differentially expressed genes, we analyzed structural changes in chromatin interactions near the differentially expressed genes between Capan-1 and PANC-1 cells. Specifically, we examined the proportion of differentially expressed genes located within genomic regions enriched for significant chromatin interaction changes and enhancer activity differences. Regions with overlapping chromatin interaction changes and enhancer activity were more likely to contain differentially expressed genes, and statistical testing confirmed the significance of this observation (see Fig. 5d; *p* value: 0.0027 by permutation test). Differentially expressed genes were further ranked based on chromatin interaction changes, and GSEA identified GO terms associated with cell motility, metastasis, and apoptotic processes (Fig. 5e).

These findings validate EpiVerse's capacity for high-fidelity replication of experimental results and underscores its value in detailed epigenetic modeling and the study of chromatin dynamics. Furthermore, the successful application to pancreatic cancer cell studies highlights EpiVerse's potential for broad applicability in various perturbation-based studies across cancers, diseases, conditions, treatments, and time points.

## Discussion

EpiVerse introduces a novel three-stage pipeline, enhancing chromatin structure prediction by integrating a wide array of imputed epigenetic signals. This approach, leveraging the Avocado model for epigenetic signal imputation, minimizes information loss and improves cross-cell-type Hi-C data imputations. The second stage employs HiConformer and its diagonal extraction algorithm, refining long-range Hi-C interaction predictions. This model, by providing multi-task Hi-C contact maps and ChromHMM states predictions, enables detailed biological annotation of chromatin interactions. MIRNet, the final stage, denoises Hi-C data, ensuring the generation of high-quality matrices. Together, EpiVerse results in the "virtual epigenetic landscape", facilitating in-depth exploration of the epigenome-chromatin structure interplay.

EpiVerse's ability to predict cell type-specific chromatin structures outperformed existing models in all performance metrics, setting it apart from other models. It integrates diverse epigenetic data, unlike models constrained by limited signals or solely DNA sequence information, allowing for a more nuanced capture of the genome's regulatory landscape. Ablation studies underscore the importance of this diverse data inclusion for accurate predictions, highlighting EpiVerse's effectiveness across different cell types and its utility in annotating previously uncharacterized tissues.

Moreover, EpiVerse excels in identifying cell type-specific regulatory elements, offering insights into tissue-specific chromatin structures and their impact on gene regulation. It differentiates tissue-specific genes between GM12878 and K562 cells using IG, shedding light on the role of epigenetic signals in gene regulation. EpiVerse also provides a deeper understanding of enhancer activity and its relationship with chromatin structure, contributing significantly to our comprehension of genomic regulation.

The construction of a Hi-C-based dendrogram for 39 tissues reveals the structured nature of chromatin interactions, highlighting regions of open chromatin in ES-cell clusters essential for pluripotency. The association between active enhancers and Hi-C interaction frequencies further illustrates the link between chromatin structure and enhancer activity. EpiVerse's analysis, including promoter hubness and its correlation with gene expression, alongside the use of GSEA, reinforces the value of enhancers in identifying tissue-specific gene expression patterns, offering comprehensive insights into chromatin architecture's role in gene regulation.

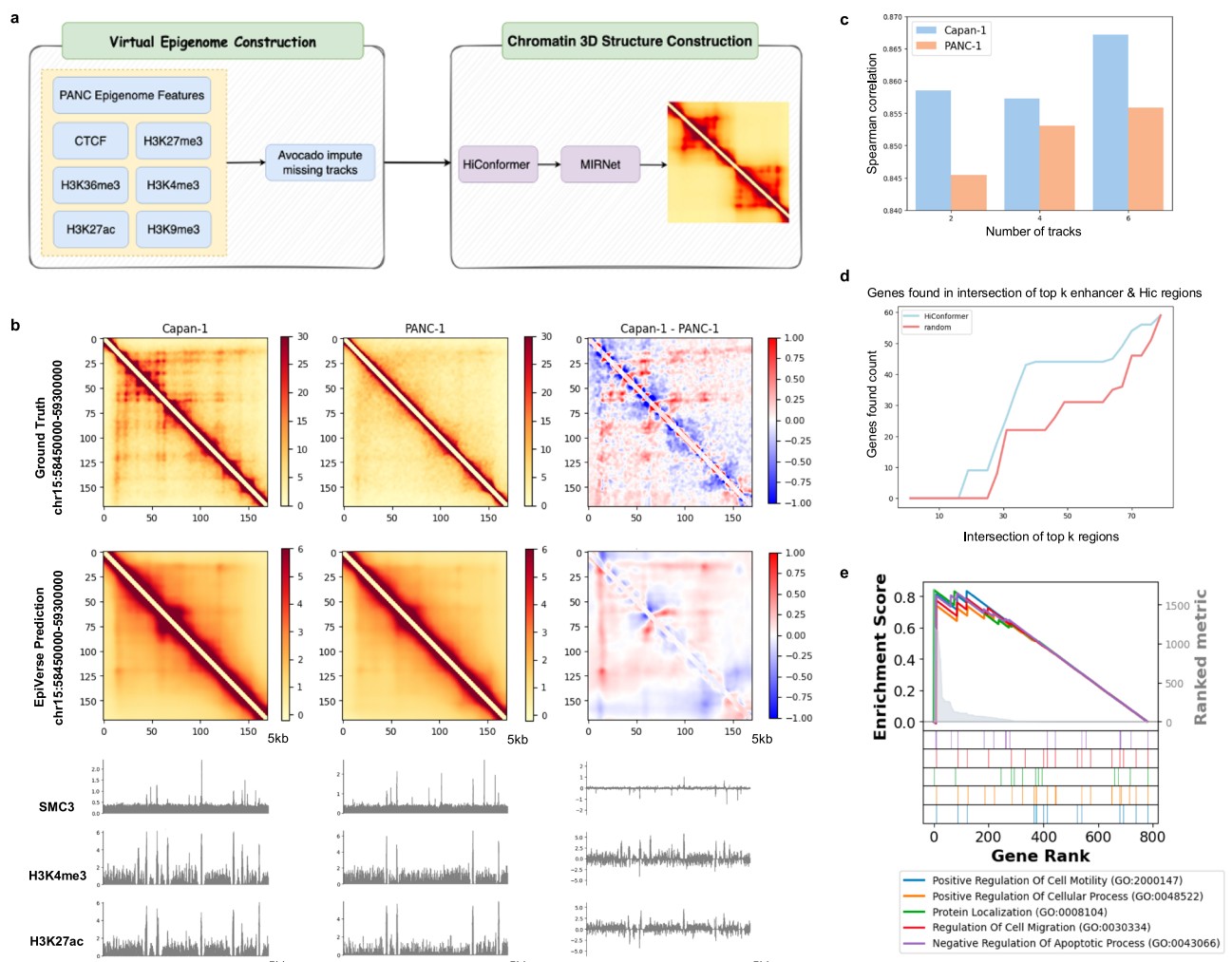

**Fig. 5 | In silico perturbation-based Hi-C experiment on pancreatic cancer metastasis. a** EpiVerse pipeline for PANC experiment reproduction: Schematic diagram illustrating the EpiVerse process from constructing virtual epigenetic signals to chromatin 3D structure modeling, using various histone modifications and CTCF as inputs to accurately replicate chromatin structures. **b** Comparison between Imputation Hi-C and experimental Hi-C: Heatmaps indicating the Hi-C interaction matrices with substantial variations across chromosome 15. The top row presents the ground truth Hi-C matrices for Capan-1 and PANC, and their differential heatmap, while the bottom row displays imputed data. **c** Enhanced predictive performance with additional tracks: A bar graph comparing the Spearman correlation coefficient of EpiVerse's predictive accuracy for Capan-1 and PANC-1 datasets. The graph highlights improvements as additional epigenetic tracks are integrated: 1 track (CTCF), 2 tracks (CTCF + H3K4me3), 4 tracks (CTCF + H3K4me3 + H3K9me3 + H3K27ac), and 6 tracks (CTCF + H3K4me3 + H3K9me3 + H3K27ac + H3K27me3 + H3K36me3). **d** Proportion of differentially expressed genes in regions enriched with chromatin Interaction and enhancer activity changes: A gene identification curve showing the proportion of differentially expressed genes located in regions enriched for both top enhancer and Hi-C signal differences in Capan-1, compared against random regions. EpiVerse consistently identifies a higher proportion of relevant genes, outperforming the random baseline. **e** Enriched GO terms of differentially expressed genes ranked by chromatin Interaction changes: GSEA of the top-ranked differentially expressed genes between Capan-1 and PANC-1 cells, revealing enrichment in GO terms associated with cell motility, metastasis, and apoptotic processes.

EpiVerse's pioneering "epigenome-level" in silico perturbation-based Hi-C experiments underscore its potential for extensive genomic exploration. Its success in modeling chromatin architecture changes in pancreatic cancer cells highlights its capability to detail chromatin looping and gene regulation dynamics. This capacity to precisely model genomic alterations and elucidate their impact on disease processes, exemplified by studies on pancreatic metastasis, underscores EpiVerse's wide-ranging potential. It suggests the tool's applicability in diverse perturbation-based studies, encompassing a variety of cancers, diseases, conditions, treatments, and temporal changes.

EpiVerse represents a noteworthy advancement in studying chromatin dynamics, yet it acknowledges specific methodological constraints. Particularly, its reliance on imputed epigenetic signals, in the absence of direct measurements, introduces a reliance that could potentially bias and inaccurately predict chromatin structures, especially when the initial signals for imputation are limited. Moreover, the exclusion of DNA methylation data from the Avocado model, due to its suboptimal imputation performance[16], signifies an important consideration. This decision was driven by concerns over integrating low-quality imputations of DNA methylation into EpiVerse, given that poor imputation suggests DNA methylation offers unique information not inferable from other epigenetic signals. Consequently, its omission points to DNA methylation's potential to contribute valuable insights into chromatin structure, underscoring the need for future studies to explore methods for accurately incorporating DNA methylation data. To overcome these limitations and enhance the virtual epigenome's accuracy, subsequent efforts should aim to include additional epigenetic tracks, like DNA methylation, not currently covered by the Avocado model, thus broadening the understanding and modeling of chromatin architecture.

Notably, discrepancies were observed between ChromHMM state predictions and the ChromHMM ground truth, a finding that was

somewhat unexpected given that the ground truth is also derived computationally from epigenetic signals. These discrepancies might be attributed to the multi-label classification approach used within a 5 kb resolution Hi-C bin (see Methods), which can lead to confusion in regions exhibiting contradictory states. This complexity hints that the relationship between epigenetic signals and ChromHMM annotations is nuanced, with the collapsing of these signals potentially obscuring the sequential relationship necessary to delineate their boundaries and order. Such insights suggest that our methodology may benefit from further refinement to accurately capture these relationships.

The study also faced challenges related to substantial GPU memory requirements for processing a large set of epigenetic signals, limiting the model's ability to predict over extensive regions (Supplementary Data 8). However, the diagonal extraction algorithm implemented in HiConformer offers a potential pathway to overcome this limitation, indicating that adjustments to target diagonals could enable predictions across larger regions in future model iterations.

These limitations identify critical areas where EpiVerse can improve, highlighting opportunities to enhance its effectiveness in chromatin structure analysis. Addressing these concerns will not only refine EpiVerse's predictive accuracy but also extend its applicability in epigenetic research, thereby contributing to a deeper understanding of chromatin dynamics and regulation.

## Methods

### Avocado training and prediction

The Avocado model has facilitated the imputation of epigenetic signals across nearly 200 tissues, with the imputation epigenetic signals dataset being made available through the ENCODE consortium. This pre-imputed data eliminates the need for additional Avocado training and prediction for tissues already included in this dataset. For tissues not covered by the existing Avocado imputations, we fine-tuned it using the latest version of the Avocado codebase, avocado-epigenome v0.3.6, along with the provided pre-trained weights for each chromosome. This process involves updating Avocado to incorporate new cell types. To integrate new cell types into the Avocado framework, we initiate the process by running the command Avocado.fit_celltypes ("available_tracks", n_epochs = 50). This function freezes the pre-trained model weights while solely adjusting the embeddings for the new cell types based on available epigenetic tracks. Subsequently, for prediction, we use Avocado.predict("tracks") with "tracks" representing the epigenetic assays not previously included in the dataset.

### Diagonal extraction algorithm

The diagonal extraction algorithm is designed to expanding model's receptive field in the chromatin structure prediction task. Unlike traditional approaches (Bin-to-Bin), where each Hi-C contact is predicted using only the features at that specific contact point, this method leverages information from surrounding regions (Area-to-Area) (Supplementary Fig. 18). By extracting entire diagonals from the ground truth Hi-C matrices, the algorithm enables each feature to incorporate neighboring information, significantly improving the predictive capacity. The algorithm extracts the ground truth Hi-C diagonals and corresponding features using the ground truth Hi-C matrices' coordinate (Fig. 1b). It focuses on the area extending from the center of diagonal to 100 bins away, ensuring a comprehensive capture of relevant genomic interactions. To optimize the quality of sampling diagonals, we set some rules to guide the selection of diagonals. The first step in this selection process involves excluding any diagonals that contain either zero values or NaN values. This precaution is necessary to prevent training instability that might arise from such incomplete or invalid data points. Following this, the algorithm implements a strategic sampling approach wherein diagonals containing peaks and those without peaks are sampled at a ratio of 1:2.5. This methodical approach to sampling ensures a balanced representation of different

chromatin states within the training data, thereby facilitating a more accurate and stable learning process for the model. These considerations in the design of the diagonal extraction algorithm significantly contributes to the robustness and improve the HiConformer performance.

### HiConformer model architecture

HiConformer model is developed with the TensorFlow framework. The HiConformer model is a multi-task CNN-Transformer architecture for accurate chromatin state and structure prediction. The initial components of HiConformer are the sequence encoder and epigenetic encoder. The sequence encoder, consisting of five CNN blocks, is tailored for the effective extraction of DNA sequence information. Each CNN block adheres to a uniform structural pattern, beginning with a Conv2D layer, followed by batch normalization and Gaussian error linear unit (GELU) activation, and concluding with max pooling. The Conv2D layers are configured as follows: the first layer with a $1 \times 5$ kernel and 64 filters; the second and third layers, each with a $1 \times 5$ kernel and 128 filters; the fourth layer with a $1 \times 5$ kernel and 256 filters; and the fifth layer with a $1 \times 2$ kernel and 256 filters. Similarly, the epigenetic encoder is also composed of five CNN blocks, designed for the extraction of epigenetic information. Its structure mirrors that of the sequence encoder, starting with a Conv2D layer, proceeding with batch normalization and GELU activation, and concluding with max pooling. The Conv2D layers in the epigenetic encoder are as follows: the first layer has a $1 \times 5$ kernel with 64 filters, the second layer also has a $1 \times 5$ kernel, but with 128 filters, the third layer utilizes a $1 \times 4$ kernel with 128 filters, the fourth layer employs a $1 \times 2$ kernel with 256 filters, and the fifth layer features a $1 \times 1$ kernel, also with 256 filters. The output from sequence and epigenetic encoder will be concatenated as a "genomic embedding" and passed to the Hi-C prediction task branch and ChromHMM prediction task branch.

The second component of HiConformer is the BERT block which is responsible for Hi-C prediction. The BERT block operates by receiving "genomic embeddings" as input, which have been processed by the earlier components. By utilizing BERT's attention mechanism, HiConformer learns complex feature interactions within the genomic embeddings. Following the BERT's attention-based processing, the output embeddings from BERT's last layer are then passed into a multi-layer perceptron (MLP). This MLP is configured with three layers: the initial layer with 512 units, followed by a second layer comprising 256 units, and a third layer, also with 256 units. The third component of HiConformer is the ChromHMM predictor, dedicated to predicting ChromHMM states. This predictor is structured as an MLP, comprising three layers. The configuration of three layers is designed in a sequential format: the first layer contains 512 units, followed by the second layer with 256 units, and the third layer also with 256 units. Similar to the BERT block, the ChromHMM predictor receives "genomic embeddings" as input. It processes these embeddings to predict the chromatin state for each Hi-C bin.

### HiConformer training and predication

In the data collection and preprocessing phase for the HiConformer model, a key focus was curating a reliable ground truth. High-resolution Hi-C data, with a resolution of 5000 base pairs per bin, were selectively procured from the 3DIV database. Concurrently, a 25-state ChromHMM dataset was sourced from the ROADMAP Epigenomics Project to serve as the ChromHMM ground truth. To streamline the model training, a state merging strategy was implemented for ChromHMM states exhibiting analogous characteristics. This included the amalgamation of Promoter Upstream (PromU), Promoter Downstream 1 (PromD1), and Promoter Downstream 2 (PromD2) states into a singular unified state. Similarly, the states of Transcription at 5' (Tx5), Transcription (Tx), and Transcription at 3' (Tx3), along with Transcription Regulation (TxReg), Transcription Enhancer at 5' (TxEnh5),

Transcription Enhancer at 3' (TxEnh3), and Weak Transcription Enhancer (TxEnhW) were consolidated. Additionally, Active Enhancer 1 (EnhA1), Active Enhancer 2 (EnhA2), and Forward Active Enhancer (EnhAF) states were merged into one state, and Weak Enhancer 1 (EnhW1), Weak Enhancer 2 (EnhW2), and Acetylated Enhancer (EnhAc) into another. The remaining states were retained in their original form. This strategic condensation resulted in a refined ChromHMM dataset encompassing 13 distinct states. For integration with the Hi-C data, the ChromHMM data underwent additional processing. Each state in the 13-state ChromHMM was encoded into a binary multi-label vector, representing the chromatin states for each 5 kb bin. This encoding was meticulously aligned with the coordinates of the Hi-C matrices from the 3DIV dataset.

In preparing the inputs for the HiConformer model, one-hot encoded DNA sequences were utilized, alongside z-score transformed epigenetic signals imputed using the Avocado model. This combination of high-quality Hi-C data, strategically simplified ChromHMM states, one-hot encoded DNA sequence, Avocado epigenetic signals are all aligned to the GRCh38 human genome assembly. This provides a solid foundation for the HiConformer model to accurately predict chromatin states and structures.

In the HiConformer training phase, we employed the diagonal extraction algorithm to prepare the model's input and training target. HiConformer then trained with a hybrid loss function in conjunction with an early-stopper mechanism and a warmup strategy to optimize the alignment between the model's predictions and the ground truth. The total training epochs is set to 5000 epochs. The Early-Stopper mechanism uses a criterion of 2000 epochs. If the model does not show improvement in the Pearson correlation coefficient criterion on the validation set for 2000 epochs, the training process will be halted. The first 30 epochs of the training process are designated as a warmup period, during which a small learning rate of 0.0000001 is applied. This gradual approach allows the model to initially adapt to the training data without making abrupt adjustments that could lead to suboptimal learning paths. After the completion of the warmup period, the learning rate is increased to 0.00001, allowing the model to more aggressively optimize its parameters.

The hybrid loss function is composed of Hi-C loss and ChromHMM multi-label classification loss, where the Total Loss is calculated as:

$$TotalLoss = HiCLoss + ChromHMMLoss \tag{1}$$

The Hi-C Loss is an amalgamation of weighted L1 and L2 losses, along with Pearson correlation coefficient and distance-stratified correlation. It's defined as:

$$HiCLoss = \frac{1}{N}\sum_{i=1}^{N}\left(W_{l1}\cdot \text{weighted\_L1}\left(y_{\text{true}_i}, y_{\text{pred}_i}\right) + (1 - W_{l1})\cdot \text{weighted\_L2}\left(y_{\text{true}_i}, y_{\text{pred}_i}\right)\right)$$
$$+ W_{\text{lwcorr}}\cdot (1 - weighted\_corr) + W_{\text{lwdcorr}}\cdot (1 - weighted\_dist\_corr) \tag{2}$$

In this equation, $N$ is the number of diagonals in a batch, $y_{\text{true}_i}$ is for the $i^{th}$ diagonal in a batch, $y_{\text{pred}_i}$ is the prediction for the $i^{th}$ diagonal from the HiConformer, $W_{l1}$ is the weighting factor between L1 and L2 loss, $W_{lwcorr}$ is the weight of weighted correlation, $W_{lwdcorr}$ is the weight of distance-stratified weighted correlation. We set parameters $W_{l1}$ to 0.1 and $W_{lwcorr}$ to 5 and $W_{lwdcorr}$ to 100. The weighted_L1, weighted_L2, weighted_corr, and weighted_dist_corr functions incorporate a distance weight, highlighting the significance of each Hi-C interaction from the 3DIV dataset.

The weighted L1 loss function, a key component of the Hi-C Loss, is defined as:

$$\text{weighted\_L1}\left(y_{\text{true}}, y_{\text{pred}}\right) = \sum_{i=1}^{N}\sum_{j=1}^{100} weight_{ij} \times \left| y_{\text{true}_{ij}} - y_{\text{pred}_{ij}} \right| \tag{3}$$

Here, each Hi-C diagonal contains 100 values, and $weight_{ij}$ refers to the weight for the $j^{th}$ value in the $i^{th}$ diagonal.

Similarly, the weighted L2 loss function, closely related to the L1 function, is expressed as:

$$\text{weighted\_L2}\left(y_{\text{true}}, y_{\text{pred}}\right) = \sum_{i=1}^{N}\sum_{j=1}^{100} weight_{ij} \times \left(y_{\text{true}_{ij}} - y_{\text{pred}_{ij}}\right)^2 \tag{4}$$

The calculation of the weighted Pearson correlation coefficient is an integral part of Hi-C Loss, calculated as:

$$r = \frac{\sum_{i=1}^{N}\sum_{j=1}^{100} weight_{ij} \times (x_{ij} - \bar{x}_i) \times (y_{ij} - \bar{y}_i)}{\sqrt{\sum_{i=1}^{N}\sum_{j=1}^{100} weight_{ij} \times (x_{ij} - \bar{x}_i)^2 \times \sum_{i=1}^{N}\sum_{j=1}^{100} weight_{ij} \times (y_{ij} - \bar{y}_i)^2}} \tag{5}$$

Here, $\bar{x}_i$ and $\bar{y}_i$ are the weighted means for each diagonal.

Additionally, the weighted distance-stratified correlation, an extension of the Pearson correlation coefficient, is defined as:

$$r_{\text{dist}} = \frac{\sum_{i=1}^{N}\sum_{j=1}^{100} weight_{ij} \times (x_{ij} - \bar{x}_i) \times (y_{ij} - \bar{y}_i)}{\sqrt{\sum_{i=1}^{N}\sum_{j=1}^{100} weight_{ij} \times (x_{ij} - \bar{x}_i)^2 \times \sum_{i=1}^{N}\sum_{j=1}^{100} weight_{ij} \times (y_{ij} - \bar{y}_i)^2}} \tag{6}$$

For the ChromHMM loss, we implemented the focal loss, which is particularly suitable for multi-label classification tasks. The ChromHMM loss is articulated as follows:

$$ChromHMMLoss = -\frac{1}{N}\sum_{i=1}^{N}\sum_{j=1}^{100}\sum_{k=1}^{13}\Big[\alpha \cdot \left(1 - y_{\text{pred}_{ijk}}\right)^\gamma \cdot \log\left(y_{\text{pred}_{ijk}}\right) \cdot y_{\text{true}_{ijk}}$$
$$+ \left(y_{\text{pred}_{ijk}}\right)^\gamma \cdot \log\left(1 - y_{\text{pred}_{ijk}}\right) \cdot \left(1 - y_{\text{true}_{ijk}}\right)\Big] \tag{7}$$

In this formula, $y_{true}$ and $y_{pred}$ represents the true and predicted values, respectively, for each of the 13 ChromHMM states, $N$ is the total number of Hi-C diagonals in a batch, $\alpha$ is constant that balances the importance of positive and negative samples, which is calculated from each state's inverse of appearance counts in whole-genome, $\gamma$ is the focusing parameter that reduces the weight of easy cases and puts more focus on hard, misclassified cases which we set as 3.5. The multi-label focal loss function computes the sum of the focal loss for each label across all 100 values in each Hi-C diagonal.

In the final phase of the HiConformer prediction process, HiConformer is implemented to prepare data for downstream MIRNet training and prediction. At this stage, the HiConformer model's weights are frozen and deployed to impute Hi-C diagonals across the entire genome. This involves systematically sliding through the genome and using the model to predict the Hi-C diagonal for each region.

## MIRNet training and prediction

In the data processing phase for MIRNet, an assembly and extraction method are employed to prepare both the input data and the corresponding ground truth. This process begins with the assembly of Hi-C diagonals into complete Hi-C matrices for each chromosome. To streamline this process, we employed the FAN-C tool[43], which facilitated efficient and effective assembly of these complex data

structures. For every chromosome, 1 Mb matrices are constructed by sliding through the chromosome with a 0.5 Mb overlap, thus creating overlapping windows that cover the entire length of the chromosome. Notably, the first and last 2 Mb regions of each chromosome are excluded from this process to avoid all zero or incomplete value regions. This results in a total of 9966 matrices, each representing a distinct 1 Mb region along the chromosomes. Parallel to the assembly of input matrices, the ground truth data for MIRNet is derived from the 3DIV Hi-C dataset. This involves extracting Hi-C matrices corresponding to the same chromosomal coordinates as those used in assembling the input matrices.

In the training phase for MIRNet, specific modifications are employed to adapt the model to the Hi-C data. The architecture retains the foundational structure of the original MIRNet but with a slight adjustment: both the number of residual recursive groups and multi-resolution blocks are set to 2. The training epochs are set for a total of 200 epochs. To prevent overfitting and ensure efficient training, an early-stopper mechanism is integrated. This mechanism is designed to halt the training process if there is no observed improvement in the validation diagonal correlation loss for 20 epochs. Another significant alteration is made to the model's loss function to better align with the characteristics of Hi-C data. The conventional MIRNet loss function is replaced with a surrogate loss, which merges the Charbonnier loss and a diagonal correlation loss. The surrogate loss is defined as:

$$\text{surrogate\_loss}(y_{\text{true}}, y_{\text{pred}}) = \text{Charbonnier\_loss}(y_{\text{true}}, y_{\text{pred}}) + \text{diag\_corr\_loss}(y_{\text{true}}, y_{\text{pred}})$$

The total loss aggregates the Charbonnier loss and the diagonal correlation loss. The Charbonnier loss is formulated as:

$$\text{Charbonnier\_loss}\left(y_{\text{true}}, y_{\text{pred}}\right) = \frac{1}{N} \sum_{i=1}^{N} \sqrt{\left(y_{\text{true}_i} - y_{\text{pred}_i}\right)^2 + \epsilon^2} \quad (8)$$

Here, $\epsilon$ is a small constant for numerical stability, we set to 1e-3 and $N$ is the total number of samples. The diagonal correlation loss is given by:

$$\text{diag\_corr\_loss}\left(y_{\text{true}}, y_{\text{pred}}\right) = 1 - \frac{1}{195} \sum_{i=4}^{198} \frac{\text{cov}\left(\text{diag}(y_{\text{true}}, i), \text{diag}\left(y_{\text{pred}}, i\right)\right)}{\text{std}\left(\text{diag}(y_{\text{true}}, i)\right) \cdot \text{std}\left(\text{diag}\left(y_{\text{pred}}, i\right)\right) + \epsilon} \quad (9)$$

This function computes the correlation between corresponding diagonals of the true and predicted matrices, focusing on diagonals from the 4th to the 198th. The diagonal correlation loss evaluates the distance-stratified relationships within the Hi-C data. The Charbonnier loss, a robust loss function, ensures that the model remains less sensitive to outliers. The overall surrogate loss synergizes these components, harmonizing the accuracy of predictions with the structural integrity of the predicted Hi-C matrices.

In the final prediction phase of MIRNet, MIRNet is adopted for genome-wide imputation. The model is tasked with the imputation of 1 Mb regions across the entire genome. To ensure thorough coverage between genomic regions, MIRNet employs a sliding window technique with a 0.25 Mb overlap. During this phase, the weights of the MIRNet model are fixed.

### Evaluation of EpiVerse and other state-of-the-art models

In our comparative evaluation of EpiVerse alongside other leading models in the chromatin structure task, we employed a standardized approach for fairness and consistency. The 3DIV IMR90 Hi-C dataset was selected as the ground truth across all models, providing a uniform foundation for model training. Additionally, Avocado epigenetic

signals were used as inputs for models requiring histone marks or TF-binding profiles.

For the Hi-C-Reg model, which predicts Hi-C contacts using epigenetic marks and TF binding profiles, we incorporated a comprehensive set of histone marks and TF binding profiles in the training. These included H3K27ac, H3K27me3, H3K36me3, H3K4me1, H3K4me3, H3K79me2, H3K9ac, H4K20me1, H3K9me3, GTF2F1, CTCF, RAD21, and DNase, in accordance with Hi-C-Reg's settings on GitHub repository[44]. Due to HiC-Reg's substantial memory requirements, which preclude the use of a whole-genome dataset, our training was confined to chromosome 1. For C.Origami, we utilized the pre-trained model weights and data available from C.Origami's publicly accessible Zenodo repository (v0.2). Regarding the Orca model, we utilized pre-trained model weights for the human fibroblast (HFF) variant, sourced from Orca's publicly available repository on GitHub[44]. This choice was motivated by the high similarity between HFF and IMR90 cells, aligning with the approach taken by C.Origami in their evaluation.

In the final evaluation phase, we first conducted whole-genome predictions across GM12878 for cross-cell type analysis. This involved analyzing each 1 Mb region predictions of the genome, with an overlap of 0.5 Mb regions, while excluding the first and last 2 Mb regions of each chromosome using various state-of-the-art models that were specifically trained on IMR90 data, including HiC-Reg, C.Origami, and Orca. Due to the inherent differences in the output resolutions and target metrics of these models, we implemented a series of processing steps to align and standardize the outputs for comparison. HiC-Reg, which requires no additional adjustments, outputs predictions directly compatible with our evaluation criteria. On the other hand, C.Origami's output, a 256 × 256 matrix representing a 2,097,132 × 2,097,132 bp region, necessitated further processing. We sliced this matrix to 244 × 244 to approximate a 2 Mb region and then applied an exponential transform to revert the original log-transformed Hi-C values. Subsequently, the matrix was resized to 400 × 400 resolution (equivalent to 5 kb/bin) using skimage's resize function, followed by distance-stratified normalization proposed by C.Origami, which normalizes each diagonal based on its mean and standard deviation to align with the ground truth. We then focused specifically on the central 1 Mb area (200 × 200 matrix). Orca's predictions also required adjustments. First, its 1 Mb module output, originally at 4 kb resolution, was realigned to a 5 kb resolution using skimage's resize function. Second, Orca predicts observed/expected (O/E) correlation values, which we normalized to match the ground truth, applying distance-stratified normalization similarly to C.Origami's approach. The final aligned predictions for each model are compared using the comprehensive metrics: Pearson correlation coefficient, Spearman correlation coefficient, distance-stratified correlation, O/E correlation and SSIM. Alongside cross-cell type analyses, we conducted intra-tissue prediction experiments to assess and compare the performance of EpiVerse with other leading models. This involved focusing on chromosomes that were specifically excluded from the training and validation sets, as per the unique design of each model.

### Evaluation metrics

**Pearson correlation coefficient** evaluates the linear relationship between predicted and ground truth Hi-C contact maps:

$$r = \frac{\sum_{i=1}^{n}(X_i - \bar{X})(Y_i - \bar{Y})}{\sqrt{\sum_{i=1}^{n}(X_i - \bar{X})^2}\sqrt{\sum_{i=1}^{n}(Y_i - \bar{Y})^2}} \quad (10)$$

where $X_i$ and $Y_i$ are the predicted and ground truth Hi-C contact values. $\bar{X}$ and $\bar{Y}$ are the mean values of X and Y., n

**Spearman correlation coefficient** measures the rank-based relationship between two Hi-C matrices. The formula is:

$$\rho = \frac{\sum_{i=1}^{n}\left(\text{rank}(X_i) - \overline{\text{rank}(X)}\right)\left(\text{rank}(Y_i) - \overline{\text{rank}(Y)}\right)}{\sqrt{\sum_{i=1}^{n}\left(\text{rank}(X_i) - \overline{\text{rank}(X)}\right)^2}\sqrt{\sum_{i=1}^{n}\left(\text{rank}(Y_i) - \overline{\text{rank}(Y)}\right)^2}} \tag{11}$$

where $rank(X_i)$ and $\backslash rank(Y_i)$ are the ranks of the predicted and ground truth values. $\overline{(rank(X))}$ and $\overline{(rank(Y))}$ are the mean ranks.

**Distance-stratified correlation** calculates Pearson correlation coefficient for each diagonal at increasing distances from the main diagonal.

$$r_d = \frac{\sum_{i=1}^{n}(X_{i,d} - \bar{X}_d)(Y_{i,d} - \bar{Y}_d)}{\sqrt{\sum_{i=1}^{n}(X_{i,d} - \bar{X}_d)^2}\sqrt{\sum_{i=1}^{n}(Y_{i,d} - \bar{Y}_d)^2}} \tag{12}$$

where $X_{i,d}$ and $Y_{i,d}$ are the predicted and ground truth values of the diagonal at distance $d$. $\bar{X}_d$ and $\bar{Y}_d$ are the means of $X_d$ and $Y_d$, respectively.

**Observed/expected (O/E) correlation** compares observed and expected Hi-C contact frequencies for predicted and ground truth data.

$$OE_{ij} = \frac{C_{ij}}{\bar{C}(d)} \tag{13}$$

$$r_{O/E} = \frac{\sum_{i=1}^{n}\left(OE_{i,\text{pred}} - \overline{OE}_{\text{pred}}\right)\left(OE_{i,\text{gt}} - \overline{OE}_{\text{gt}}\right)}{\sqrt{\sum_{i=1}^{n}\left(OE_{i,\text{pred}} - \overline{OE}_{\text{pred}}\right)^2}\sqrt{\sum_{i=1}^{n}\left(OE_{i,\text{gt}} - \overline{OE}_{\text{gt}}\right)^2}} \tag{14}$$

where $C_{ij}$ is the observed contact frequency between loci $i$ and $j$. $\bar{(C)}(d)$ is the expected contact frequency at distance d. $OE_{i,pred}$ and $OE_{i,gt}$ are the observed / expected values for the predicted and ground truth $i_{th}$ diagonal. $n$ represents the total number of valid diagonals used in the calculation across the Hi-C matrices.

**Structural similarity index (SSIM)** evaluates the structural similarity between two Hi-C matrices.

$$\text{SSIM}(X, Y) = \frac{(2\mu_X\mu_Y + C_1)(2\sigma_{XY} + C_2)}{(\mu_X^2 + \mu_Y^2 + C_1)(\sigma_X^2 + \sigma_Y^2 + C_2)} \tag{15}$$

where $\mu_X$ and $\mu_Y$ are the mean values of the predicted and ground truth matrices. $\sigma_X^2$ and $\sigma_Y^2$ are their variances. $\sigma_{XY}$ is the covariance between the two matrices. $C_1$ and $C_2$ are constants to avoid division by zero.

## DNA sequence motif discovery

We explored motifs associated with chromatin structure using HiConformer's sequence encoder. Initially, DNA sequences were transformed into one-hot encoded formats. These sequences were then processed through HiConformer's first CNN block, which is equipped with 64 kernels, for generating activation values. For each of these 64 kernels, we focused on identifying bases where the activation values exceeded a threshold of 2. After pinpointing these bases, they were extended to include a range of 15 base pairs, spanning from −7 to +7 bases around the activation site. The final step involved calculating the frequency of each nucleotide—adenine (A), thymine (T), cytosine (C), and guanine (G)—within these extended sequences for each kernel. This meticulous process resulted in the generation of 64 distinct PWMs, each corresponding to one of the kernels. Each of these PWMs, was then inputted into TOMTOM version 5.5.5.[45] to match relevant motifs for each of the 64 kernels.

## Assessing epigenetic influence on chromatin with IG

IG was employed to assess the significance of epigenetic signals in chromatin structure and to identify cell type differential genes. We implemented IG with a baseline of all-zero vectors and used 64 interpolated steps to create an integral path. This approach allowed us to compute the gradient of epigenetic signals with respect to chromatin structure, revealing their relative importance. To link gene regulation with chromatin structure via IG, we first identified the promoter region of each gene using our ChromHMM predictions' promoter state. These identified promoter regions were then extended by +0.25 Mb and −0.25 Mb to encompass a broader area. Within these extended regions, we calculated the mean IG importance for each epigenetic track. This mean importance score for each track was further averaged to derive a single importance value for each gene. The final step in our process involved ranking the genes based on their calculated importance values. This ranking provided insights into the relative significance of genes in terms of their impact to chromatin structure. Furthermore, to analyze tissue-specific differential genes, we subtracted the gene importance values of one tissue from another, laying the groundwork for further pre-ranked GSEA.

## Pre-ranked GSEA

In the pre-ranked GSEA phase of our study, we employed GSEApy version 1.0.4[46], a Python implementation of GSEA that also serves as a wrapper for Enrichr[47]. Enrichr is renowned for its extensive collection of diverse gene set libraries, making it an invaluable resource for our analysis. To explore oncogenesis signatures, we utilized the "MSigD-B_Oncogenic_Signatures" gene set library from Enrichr. This library provided us with a comprehensive collection of gene sets associated with oncogenic processes, enabling us to pinpoint key signatures related to cancer development. For the identification of cancer-related modules, we turned to the "MSigDB_Computational" library within Enrichr. This library focuses on computational gene sets, offering insights into complex cancer-related molecular interactions and pathways. For identifying tissue-specific genes, we utilized the "GTEx_Tissues_V8_2023" library from Enrichr. Additionally, to investigate biological processes, we employed the "GO_Biological_Process_2023" library from Enrichr. The gene ranking provided to the pre-ranked GSEA, as described in each respective section of our study, was tailored to the specific objectives of each analysis. By leveraging these diverse and specialized gene set libraries, our pre-ranked GSEA approach allowed us to gain nuanced insights into the roles of specific genes in oncogenesis, cancer modules, and broader biological processes.

## Enrichment analysis in Hi-C variation region

To delve into the connection between chromatin structure and regulation, we focused on the genes located within the Hi-C variation regions across different tissues. We conducted an enrichment analysis on the top 200 genes from the most variable regions using Metascape[48]. Our analysis focused on the data from MetaScape's PaGenBase[49] results, which allowed us to gain meaningful insights into the interplay between pattern genes and chromatin structure. This approach was instrumental in enhancing our understanding of how chromatin structure influences gene regulation across different tissue types.

## Active enhancer and region importance ranking

We utilized the predictions ChromHMM active enhancer states from HiConformer model to identify tissue-specific genes, capitalizing on our earlier finding that enhancers are both highly tissue-specific and closely tied to gene regulation. The process began with the identification of promoter regions for each gene using our ChromHMM predictions' promoter state. These identified promoter regions were then extended by +0.25 Mb and −0.25 Mb to ensure a comprehensive

analysis that included the surrounding regulatory landscape, which could significantly influence gene expression. The importance score for each gene was assigned based on the total count of active enhancers within these extended regions. Furthermore, whenever assessing the importance of genomic regions on a broader scale, we calculated the importance of each 1 Mb region across the entire genome. This was done by summing the counts of active enhancers within these regions, providing a quantifiable measure of regulatory activity across different genomic segments.

## Hi-C-based dendrogram construction

EpiVerse has been utilized to impute Hi-C data for 39 representative human tissues, providing a comprehensive dataset for analyzing chromatin structure relationships between different tissues. To achieve this analysis, we first constructed a Hi-C-based dendrogram, focusing on the genomic structure and similarities across these tissues. The process involved iterating through each 1 Mb region of the whole genome for all the tissues. For each 1 Mb region, we compared the tissues' 1 Mb region matrix against the mean matrix, calculating from all tissues' 1 Mb region matrix. This comparison was done by subtracting the mean matrix from the tissues' matrix and then calculating the L1 distance between the tissues' matrix and the mean matrix. The result of this calculation gave us a feature vector for each tissue, specifically representing the chromatin structure distance in that 1 Mb genomic region.

Using these feature vectors for each tissue and genomic region, we proceeded to perform hierarchical clustering. This method groups tissues based on the similarity of their feature vectors, thus revealing the hierarchical relationships in chromatin structure across different tissues. By visualizing these relationships in a dendrogram, we gain valuable insights into the complex dynamics of chromatin organization across different human tissues.

## Virtual 4C promoter hubness calculation

Promoter hubness is a metric that quantifies the extent of genomic interactions involving gene promoters. To calculate this metric, we utilized ChromHMM predictions promoter states from HiConformer to identify the location of each gene's promoter. Once these promoter locations were pinpointed, we proceeded to analyze their interactions with other genomic regions, focusing specifically on the areas within ±0.25 Mb of the promoter. This analysis involved calculating the frequency of interactions each promoter region had interacted, based on imputed Hi-C interaction data. The average number of these interactions was then computed to determine the promoter hubness score for each gene. This score reflects the level of connectivity a gene's promoter has within the genomic network, with a higher promoter hubness score indicating a promoter that is part of a dense network of interactions. Such a promoter is potentially more influential in gene regulation and chromatin architecture, highlighting its significance in the genomic interaction landscape.

## Analyzing Hi-C interaction changes in gene regulation

Hi-C interaction differences between the two cell lines were calculated for each 5 kb bin by considering contact counts within a ±15-bin window associated with each region. For each bin, significant interaction changes, defined as exceeding thresholds of more than 3 or less than −3 normalized contact count differences, were assigned a binary value of 1, while all other changes were marked as 0. The values of all bins within the associating window were summed to compute a summary Hi-C interaction change score for each region. The highest possible scores for a region were 28, since the predicted Hi-C contact map does not consider the immediate vicinity (±1 bin) of a region.

Regions were independently ranked based on their Hi-C interaction scores and active enhancer differences, with the latter determined by the differential count of bins exhibiting the active enhancer state. The intersection of these ranked lists was used to evaluate the number of differentially expressed genes located in regions enriched for both Hi-C interaction and enhancer changes. To account for potential regulatory influences, gene boundaries were extended by ±0.6 Mb during the analysis. The statistical significance of the overlap was assessed through a permutation test, in which regions were randomly shuffled 10,000 times to calculate a $p$ value for the observed associations.

Differentially expressed genes between Capan-1 and PANC-1 cells were mapped to genomic regions at a 1 Mb resolution. The Hi-C interaction change score of all bins within an ±0.6 Mb range of each gene was summed and then used to rank genes. This ranked list served as input for GSEA to identify GO terms under the biological process ontology associated with the top differentially expressed genes.

## Reporting summary

Further information on research design is available in the Nature Portfolio Reporting Summary linked to this article.

## Data availability

The GRCh38/hg38 reference genome used in this study was downloaded from the UCSC database https://hgdownload.soe.ucsc.edu/goldenPath/hg38/bigZips. The Hi-C data for IMR90, K562, and GM12878 cell lines with the MboI restriction enzyme used for training models have been obtained from the 3DIV database http://3div.kr/download. The ChromHMM data for IMR90 (E017), GM12878 (E116), and K562 (E123), based on the 25-state, 12-mark chromatin state model from the ROADMAP Epigenomics project, were also used for training models and are available at the ROADMAP Epigenomics portal https://egg2.wustl.edu/roadmap/web_portal/index.html. High-quality imputed epigenetic signals from Avocado are listed in Supplementary Data 1, and the ENCODE accession IDs for each track can be found in the following repository: https://github.com/jhhung/EpiVerse/blob/main/Avocado/Avocado_metadata.csv. The In silico perturbation-based Hi-C datasets used in this study are available from the NCBI GEO database under accession codes GSE137374, GSE68976, and GSE149103. The pre-trained model weights for IMR90, K562, GM12878, and EpiVerse-imputed ChromHMM states across 41 tissues/cell lines generated in this study are available at https://zenodo.org/records/13759557.

## Code availability

The source code of EpiVerse, along with scripts for preparing and analyzing data, is available on the GitHub repository: https://github.com/jhhung/EpiVerse[44]. A web browser for visualizing the virtual epigenetic landscapes of the 39 tissues generated by EpiVerse can be found at https://epiverse.jhhlab.tw.

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

## Acknowledgements

This work was supported in part by grants from the National Science Council (107-2320-B-009-002-MY2, 110-2221-E-A49-069-MY3) to J.H.H. We extend our heartfelt thanks to Chin Yang for the initial exploration of this topic, laying the groundwork for our study. Our gratitude also goes to Chao-Hsi Lee for his invaluable contribution to constructing the early conceptual framework that guided our research. Their pioneering efforts have been instrumental in the development of this work. Additionally, we would like to thank Ching-Chun Jao and Po-Yen Wang for their

valuable assistance during the revision process, which enhanced the quality of this manuscript.

## Author contributions

J.-H.H. initiated and led the project, contributed to manuscript writing, and supervised the project. M.-Y.L. contributed to model development, discussions, draft manuscript writing, and manuscript revision. Y.-C.L. performed model development and contributed to discussions.

## Competing interests

All authors declare no competing interests.
