## [Transparent Peer Review file · Nature Communications]

Unveiling Chromatin Dynamics with Virtual Epigenome

Corresponding Author: Professor Jui-Hung Hung

Version 0:

Reviewer comments:

Reviewer #1

(Remarks to the Author)

In this manuscript, Lin et al. introduce a novel deep-learning framework called EpiVerse, designed to predict Hi-C contacts using DNA sequences and imputed epigenetic signals as input. The framework involves three main steps: 1) imputation of 71 distinct epigenetic signals from a selected set of epigenetic profiles using Avocado; 2) application of a multi-task learning strategy to enable simultaneous predictions of ChromHMM chromatin states and Hi-C contacts; and 3) utilization of the MIRNet framework to denoise the predictions from the second step. The authors demonstrate that EpiVerse outperforms existing methods in terms of cross cell-type predictions. Furthermore, they generate chromatin contact maps for 39 human tissues and illustrate the potential of EpiVerse in facilitating in silico perturbational studies to explore the effects of genetic and epigenetic alterations on 3D genome organization.

Overall, I feel the work interesting. However, both the analysis and the writing require significant improvement, particularly in addressing the following concerns.

1) It is not surprising that incorporating more epigenetic signals into the model improves the prediction accuracy of Hi-C contacts. However, the manuscript does not clearly define the minimum input requirements for EpiVerse. This is a crucial consideration, especially in practical scenarios. For instance, in a tissue not included in the pre-trained Avocado model, how many ChIP-Seq experiments are necessary to accurately predict Hi-C contacts? Moreover, if data from a similar, but not identical, tissue is used, what degree of performance reduction should be expected?

2) The performance of C.Origami is surprisingly poor – significantly worse than that of Orca – which is contradictory to the findings reported in the original paper (DOI: 10.1038/s41587-022-01612-8). It is important to note that the authors re-trained the C.Origami models and compared their EpiVerse models to these re-trained C.Origami models. While the original C.Origami models were trained using DNA sequences, CTCF ChIP-Seq, and ATAC-Seq data, the authors replaced the ATAC-Seq data with DNase-Seq data during the model re-training. Although ATAC-Seq and DNase-Seq are generally similar in profiling chromatin accessible regions, this substitution could potentially explain the unexpectedly poor performance of C.Origami in their evaluation. To ensure a fair comparison, the authors should re-evaluate the performance of C.Origami using the officially available models and data from <https://zenodo.org/records/7226561>.

3) Additionally, upon closer examination of the Methods section, I noticed that specific post-processing steps were applied to different methods to ensure they were compared at the same resolution. However, these processes, including matrix resizing based on skimage's resize function, might have altered the value distributions of other methods, leading to an unfair comparison. What are the results of the same metrics for each method before these post-processing steps?

4) The pretrained Avocado model used by this work can impute epigenetic signals for 84 assays (<https://zenodo.org/records/4774521>); however, the authors selected only 71 of these for training the EpiVerse models. They should at least provide an explanation for why these 71 tracks were chosen and why the others were excluded.

5) Probably due to the huge list of dependent packages (<https://github.com/jhung/EpiVerse/blob/main/environments/HiConformer.yml>), I was unable to set up the HiConformer environment successfully due to frequent conflicts. As a methods paper, it is essential for the authors to simplify the installation procedure and thoroughly test their code on multiple machines to ensure it is runnable.

In addition to the major concerns mentioned above, there are numerous other issues. Specifically, many figure panels and

captions are not well-prepared and fail to effectively convey the intended information.

- 1) On page 9, the authors state that "... we retrieved comprehensive epigenetic datasets, which included 71 selected tracks of epigenetic signals (Table S1). These tracks covered a broad spectrum of epigenetic markers, including histone modifications, transcription factor (TF) binding sites, RNA expression profiles, and ATAC-seq." However, "ATAC-Seq" is missing from Table S1.
- 2) The precise definition of SSIM, an important metric used for model evaluation in this study, is not provided.
- 3) What are the values shown in Figure 2d? Are they average correlations between the predicted contact signals and the experimental signals? If so, are they Pearson correlations, Spearman correlations, or distance-stratified correlations?
- 4) In the first paragraph on page 13, the authors state, "Interestingly, we observed that the motifs identified by the sequence encoder CNN are not present in the imputed epigenetic signals." However, no figures are provided to support this statement.
- 5) Genomic coordinates should be added to all figures containing contact heatmaps to facilitate reproducibility of the analysis.
- 6) In Figure 3d, the loop shown in the center appears to be a canonical CTCF loop. It would be more informative to overlay both enhancer signals and CTCF signals on top of the Hi-C heatmaps. Additionally, there seems to be a typo in the corresponding figure caption. Based on the context, it should be the blue color in the rightmost column that indicates regions with stronger interactions in K562, rather than the red color.
- 7) In Figure S9, the caption states that "Each panel presents Hi-C contact maps for selected genomic regions, showcasing the distribution of active enhancers ..." However, only heatmaps are shown for each region, and the enhancers are not displayed.
- 8) The figure caption for Figure 4d does not align with the text that references this figure. According to the caption, this figure shows the enrichment of gene sets associated with active enhancers. However, the text suggests that this figure should demonstrate GO terms of genes in regions with the largest variation in chromatin structure.
- 9) In Figure S12, where are the promoter-based results? Based on the main text that references this figure (Page 18), one would expect two sets of analyses: one using gene promoters and another using active enhancers.
- 10) In addition to the missing results in Figure S12, what do the labels such as "70-79 Up" and "40-49 Up" mean in the figure legend?
- 11) In the main text that references Figure S15, the authors state, "This in silico perturbational Hi-C experiments by EpiVerse mirrored the results of the original study, yielding new chromatin contacts and reinforced domain boundaries, as evidenced by the congruence with real Hi-C experiment observations." Based on this description, one would expect Figure S15 to include both predicted and observed contact matrices, demonstrating their agreement. However, the real observed contact matrices are missing from this figure.
- 12) In Figure S16, the column labels are missing. Additionally, to provide a more comprehensive view of the locus and the associated dynamics, genes, RNA-Seq data, and relevant epigenetic signals should be overlaid on top of the heatmaps.
- 13) There are several issues with figure 5 alone: First, it appears that panels b and c are swapped. Second, in panel b, the epigenetic signals before and after perturbation should be displayed to help readers better understand the relationship between the dynamics of epigenetic signals and chromatin structure. Third, the caption for panel e is missing. Finally, the relevance of panels d and e is unclear, as they do not seem to be related to chromatin interactions.
- 14) On page 20, there is a lack of a literature reference for the sentence: "For the insertion experiment, we were inspired by a pivotal study that highlighted the importance of engineering chromosomal contact domains."
- 15) To ensure reproducibility, the pre-trained models for GM12878 and K562 should also be uploaded. Currently, only the model trained in IMR90 has been made available (<https://zenodo.org/records/10889119>).
- 16) In the EpiVerse browser (<https://epiverse.jhlab.tw/>), entering values for "Chromosome," "Start Location," and "Tissue Sample," as shown in Supplementary Figure 13, resulted in an error message: "Error fetching matrices: Please check your input range." Please verify whether the browser is functioning correctly.

(Remarks on code availability)

Probably due to the huge list of dependent packages

(<https://github.com/jhung/EpiVerse/blob/main/environments/HiConformer.yml>), I was unable to set up the HiConformer environment successfully due to frequent conflicts. As a methods paper, it is essential for the authors to simplify the installation procedure and thoroughly test their code on multiple machines to ensure it is runnable.

Reviewer #2

(Remarks to the Author)

The manuscript introduces EpiVerse, a novel three-stage pipeline designed to enhance chromatin structure prediction, marking a significant advancement in understanding chromatin dynamics. EpiVerse employs the Avocado model for accurate epigenetic signal imputation, improving Hi-C data across various cell types with minimal information loss and demonstrating up to 200% improvement in SSIM metrics over existing models. The second stage introduces the HiConformer model, utilizing a diagonal extraction algorithm to refine predictions of long-range Hi-C interactions, supporting multi-task predictions including Hi-C chromatin interactions and chromatin state annotation. The third stage, MIRNet, ensures data quality by denoising Hi-C matrices. EpiVerse excels in identifying cell type-specific regulatory elements and differentiating tissue-specific chromatin structures, providing insights into gene regulation and enhancer activity. Additionally, a Hi-C-based dendrogram reveals the structured nature of chromatin interactions across tissues, linking enhancers to chromatin architecture and gene expression. EpiVerse's ability to perform in silico perturbational experiments without extensive laboratory work is a valuable contribution, allowing detailed modeling of genomic alterations, as demonstrated in pancreatic cancer studies, highlighting its broad potential for exploring disease processes and other genomic inquiries.

The noteworthy results include the significant improvement in chromatin structure prediction accuracy and the ability to identify cell type-specific regulatory elements. This work is highly significant to the field and related fields, as it addresses limitations of traditional methods that rely heavily on DNA sequence information and limited epigenetic signals. By integrating diverse epigenetic data, EpiVerse offers a comprehensive understanding of chromatin structure and gene regulation, with potential impacts on related fields such as cancer research. The conclusions are well-supported by the data, with comparative analysis and ablation studies underscoring the importance of incorporating a wide range of epigenetic signals. The successful application to pancreatic cancer cell studies further supports its broad applicability.

While the reliance on imputed epigenetic signals could introduce biases, this is acknowledged, and future work should address this limitation. The methodology is robust, employing advanced machine learning techniques and a novel three-stage pipeline, although the exclusion of DNA methylation data due to suboptimal imputation performance is a noted limitation. The methods section provides sufficient detail for reproducibility, with clear descriptions of the imputation process, use of HiConformer, and generation of Position Weight Matrices. Overall, the study presents a noteworthy advancement with well-supported conclusions and sound methodology, and addressing the noted limitations in future work will enhance its impact.

Comments:

1. Some content in the Introduction overlaps with the Conclusion and Rationale sections. It's recommended to condense these parts for clarity.
2. Predicting chromatin state is a major functionality of EpiVerse. It would be beneficial to expand on this feature and its results. Providing downloadable chromatin states for various cells would offer readers an alternative to chromHMM.
3. In Fig. 1, the HiConformer model, a core component of this paper, is not described clearly enough. Adding some illustration would help readers better understand the model by combining textual description with a visual representation.
4. All heatmaps (e.g., Fig. 2a and Fig. 2d) should include a color scale legend.
5. All boxplots (e.g., Fig. 4b and Fig. S10) should include a test for statistical significance.
6. Some computational results in this paper can be verified by other literature (e.g., at the end of Page 18, Paragraph 2: This observation is...in transcription levels [33]). Clearly indicate the specific reference position or figure.
7. In Table S1, please list the specific download addresses for each data source.
8. Each table should include explanation of the terms' name.
9. Please ensure that figures associated with chromosomes or genomic regions are marked or annotated with the relevant chromosome or genomic coordinates, such as the 2D heatmaps in Fig. 2a.
10. It is recommended to cite original articles in your references. For instance, the citation for Hi-C technology on the first paragraph on page three should reference its original source: Eriz, Science, 2009.
11. On the EpiVerse browser (<https://epiverse.jhhlab.tw>), it is suggested to provide guidance and examples on the webpage as an error occurs: "Error fetching matrices: Please check your input range."
12. Fig. 2b should include a y-axis title.
13. Bar charts, such as in Fig. S7, should include error bars.
14. Page 9, Paragraph 1, the abbreviation "3DIV" needs an explanation.
15. Page 11, Paragraph 1, provide the full name and explanation for "O/E correlation."
16. Page 12, Paragraph 3, "IG" needs a citation and explanation.
17. Page 13, Paragraph 1, "PWM" needs a citation and explanation.
18. Page 13, Paragraph 2, "GSEA" needs an explanation.
19. Page 13, Paragraph 2, check "v" cases for "BMI1_DN_MEL18_DN.V1_DN. BMI1" and "PDGF_ERK_DN.v1_DN."
20. In Fig. 3c, since the data is sparse and colors are similar, label factor names directly in the figure for easier identification.
21. At the end of Fig. 3d's caption, the color description may be reversed.
22. Page 29, Paragraph 2, provide information monitoring the HiConformer training process, likewise epoch, loss, AUROC, and other diagnostic curves.
23. In Fig. 5, panel "b" and "c" reversed with their captions.
24. It is more precise to use "by up to 200% improvement" than "by a 200% improvement" in paragraph 2 of the Conclusions section.

(Remarks on code availability)

In the README on EpiVerse's GitHub, there are instructions for installing three software components. I tried to install them on my computer (ProductName: macOS; ProductVersion: 11.4; BuildVersion: 20F71). As a result, the "Avocado Environment" and the "fanc Environment" were successfully installed. However, an error occurred while installing the "HiConformer training and MIRNet training Environment," as follows:

```
EpiVerse % conda env create -f environments/HiConformer.yml
```

```
Collecting package metadata (repodata.json): - WARNING conda.models.version:get_matcher(538): Using .* with relational operator is superfluous and deprecated and will be removed in a future version of conda. Your spec was 1.7.1.*, but conda is ignoring the .* and treating it as 1.7.1
```

```
WARNING conda.models.version:get_matcher(538): Using .* with relational operator is superfluous and deprecated and will be removed in a future version of conda. Your spec was 1.9.0.*, but conda is ignoring the .* and treating it as 1.9.0
```

```
WARNING conda.models.version:get_matcher(538): Using .* with relational operator is superfluous and deprecated and will be removed in a future version of conda. Your spec was 1.6.0.*, but conda is ignoring the .* and treating it as 1.6.0
```

```
WARNING conda.models.version:get_matcher(538): Using .* with relational operator is superfluous and deprecated and will be removed in a future version of conda. Your spec was 1.8.0.*, but conda is ignoring the .* and treating it as 1.8.0  
done
```

```
Solving environment: failed
```

```
ResolvePackageNotFound:
```

```
- conda-forge/linux-64::libdeflate==1.10=h7f98852_0  
- conda-forge/linux-64::libprotobuf==3.20.1=h6239696_4  
- defaults/linux-64::tensorboard-data-server==0.6.1=py37h52d8a92_0  
- conda-forge/linux-64::xorg-libxau==1.0.11=hd590300_0  
...
```

Reviewer #3

(Remarks to the Author)

Lin et al. present a computational method called EpiVerse that bypasses reliance on limited ChIP-seq inputs by leveraging imputed epigenetic signals and advanced deep learning techniques. The results show that EpiVerse significantly improves the accuracy of cross cell-type Hi-C prediction, predicts Hi-C contact maps across an array of 39 human tissues, and facilitates unprecedented in silico perturbational experiments at the "epigenome-level". This is a good tool that can help researchers to better understand into chromatin dynamics and their implications for in cellular functions and pathological conditions. However, there are few comments and concerns that need to be addressed to strengthen the manuscript.

1. What is the difference between Figure 2 and Supplementary Figure 1? The authors clearly state what they are trying to show, including stating the cell lines that are being tested, to lessen the confusion instead of copying and pasting the same caption for both Figures. The overall captions for all the figures can be more descriptive.
2. On page 9, a brief description of "hybrid L1-Diagonal-corr loss function" would be beneficial.
3. O/E correlation is explained for the first time on page 33. The authors should introduce this when they first use the term on page 11. I.e. observed / expected (O/E) correlation (Figure 2b).
4. What was the reason behind choosing "chromosomes 1 through 18 were designated for the training set, chromosomes 19 and 20 for the validation set, and chromosomes 21 and 22 were used as the testing set"? Is it due to the decreasing chromosome sizes?
5. The authors should highlight that EpiVerse is publicly available on GitHub in the abstract.
6. What is the reproducibility of EpiVerse? Showing this will greatly benefit and strengthen the manuscript.
7. The authors mention that EpiVerse uses substantial GPU memory requirements for processing. It would be beneficial to actually mention the computational time and GPU memory needed to run this pipeline. As well as having a comparison table/figure that showcases the computational time and GPU memory from EpiVerse and the other methods that were being compared would strengthen the manuscript.

(Remarks on code availability)

README file is very well organized, well explained, and user friendly.

Version 1:

Reviewer comments:

Reviewer #1

(Remarks to the Author)

The authors have addressed most of my previous concerns, which I greatly appreciate. However, several key issues remain:
1. Regarding the concern about using scikit-image's resize function to adjust the outputs of C.Origami and Orca to match

EpiVerse's resolution, the authors explained that this approach was consistent with the original C.Origami study. However, this reasoning is not convincing. In the original C.Origami paper, the ground truth matrices were at ~8 kb resolution, while Akita predicted matrices at ~2 kb resolution and Orca at ~4 kb resolution. In that context, resizing simply coarsens the matrices to align with the lower-resolution ground truth, which is reasonable. However, in the current study, the ground truth matrices are at 5 kb resolution, and the authors resized C.Origami-predicted matrices (originally at 8 kb) to this finer resolution. A coarser resolution matrix inherently lacks the interaction details necessary for finer resolution analysis, and resizing such matrices to finer resolutions is likely to artificially degrade their performance. To ensure a fair comparison, the authors should calculate performance metrics for each method prior to resizing, especially for C.Origami. Comparisons should be made using ground truth matrices at matching resolutions (e.g., 8 kb for C.Origami and 4 kb for Orca).

Alternatively, I suggest the authors include a supplementary analysis by retraining EpiVerse models at resolutions consistent with C.Origami. This would allow direct comparisons without resizing. Addressing the resolution discrepancy properly is critical to substantiate the claim that EpiVerse outperforms C.Origami.

2. I appreciate the addition of active enhancers to Figure 3d and Supplementary Figure 10. However, I am surprised that, in some cell lines (e.g., K562 in Figure 3d and the leftmost columns for each region in Supplementary Figure 10), the entire region appears to be covered by active enhancers. Including H3K27ac signal tracks as a reference would enhance the interpretation of these regions.

3. In my previous comments, I raised concerns about Figures 5d and 5e being unrelated to chromatin interactions. In this revision, the authors explained that these panels demonstrate the ability of imputed ChromHMM data to identify tissue-specific upregulated genes. However, these analyses appear to highlight the capabilities of Avocado rather than EpiVerse, as identifying ChromHMM chromatin states is straightforward when epigenetic signals are available. A more compelling analysis would involve investigating whether regions predicted by EpiVerse to undergo dramatic 3D genomic changes are enriched for differentially expressed genes. Furthermore, Figure 5e presents a GSEA analysis of the top 200 upregulated genes marked by active enhancers in Capan-1. How do the results compare when analyzing the top 200 upregulated genes without considering active enhancers? Again, instead of focusing on differentially expressed genes, it would be more informative to determine whether dynamic changes in 3D genome structure alone can identify genes enriched in pathways related to cell motility and cancer metastasis.

4. The manuscript requires careful proofreading to ensure clarity and coherence. For example, the caption for Figure 5b includes the following sentence: "Gene Discovery via Active Enhancer Counts: Line graph demonstrating the discovery rate of genes using the active enhancer count approach, validating its effectiveness in identifying genes with differential expression." This description appears irrelevant to the figure content. Such inconsistencies should be corrected to improve the overall quality of the manuscript.

(Remarks on code availability)

The authors have done a great job updating both the code and the documentation. The software can now be installed and run successfully.

Reviewer #2

(Remarks to the Author)

The author has addressed all of our questions and concerns, and we are satisfied with the response.

(Remarks on code availability)

Yes, I was able to install and run the code successfully. The provided README file contained sufficient instructions for installation and execution, making the process straightforward. The code is well-structured and serves as a valuable resource for the community.

Reviewer #3

(Remarks to the Author)

I thank the authors for addressing all the questions and concerns that I had. They have responded very clearly and well to the questions, and I recommend accepting this article once they address some minor issues about their GitHub.

(Remarks on code availability)

The authors have significantly improved the GitHub with thorough explanations and clear instructions. The interactive browser, EpiVerse HiCViewer, is a nice touch. For HiConformer training and MIRNet training Environment and Data processing environment, it would be nice to have lines "git clone <https://github.com/jhhung/EpiVerse.git>; cd EpiVerse" before the conda env create comment to lessen the confusion. The authors should also list the recommended conda version that is needed to create the environments as I'm getting a warning message (conda 23.7.4):

```
[leeh7@lri-r04 EpiVerse]$ conda env create -f environments/HiConformer.yml
```

```
Collecting package metadata (repodata.json): - WARNING conda.models.version:get_matcher(556): Using .* with relational operator is superfluous and deprecated and will be removed in a future version of conda. Your spec was 1.9.0.*, but conda is ignoring the .* and treating it as 1.9.0
```

```
WARNING conda.models.version:get_matcher(556): Using .* with relational operator is superfluous and deprecated and will be removed in a future version of conda. Your spec was 1.6.0.*, but conda is ignoring the .* and treating it as 1.6.0
```

```
WARNING conda.models.version:get_matcher(556): Using .* with relational operator is superfluous and deprecated and will be removed in a future version of conda. Your spec was 1.8.0.*, but conda is ignoring the .* and treating it as 1.8.0
```

```
WARNING conda.models.version:get_matcher(556): Using .* with relational operator is superfluous and deprecated and will be removed in a future version of conda. Your spec was 1.7.1.*, but conda is ignoring the .* and treating it as 1.7.1
```

I was able to install all 3 environments.

Version 2:

Reviewer comments:

Reviewer #1

(Remarks to the Author)

The authors have thoroughly addressed all of my previous concerns. I have no further concerns and fully support the acceptance of this manuscript in its current form.

(Remarks on code availability)

The software can be installed and run successfully.

We would like to express our sincere gratitude to the reviewers and the editor for their dedicated time and constructive feedback. We greatly appreciate the insightful comments provided, and we have addressed each of them in the following responses.

Reviewer #1 (Remarks to the Author):

Major Concerns

- 1) It is not surprising that incorporating more epigenetic signals into the model improves the prediction accuracy of Hi-C contacts. However, the manuscript does not clearly define the minimum input requirements for EpiVerse. This is a crucial consideration, especially in practical scenarios. For instance, in a tissue not included in the pre-trained Avocado model, how many ChIP-Seq experiments are necessary to accurately predict Hi-C contacts? Moreover, if data from a similar, but not identical, tissue is used, what degree of performance reduction should be expected?

We greatly appreciate the reviewer's insightful question. We agree that omitting the minimum input requirement could lead to confusion, and we apologize for this oversight. As stated in the original Avocado paper, '*Our analysis suggests that one can achieve good quality imputations with as little as **a single track** of training data in a given biosample*', and since EpiVerse relies on Avocado for imputation, the minimum requirement is a single track to predict Hi-C contact maps.

To address the editor's concerns regarding performance on unseen or similar but not identical tissues, we conducted additional experiments using the primary pancreatic cancer cells (PANC-1) and metastatic pancreatic cancer cells (Capan-1) datasets. These datasets are particularly suitable for evaluating cross-tissue generalization. For the single-track scenario, we used the CTCF track to impute other tracks, as it is the most relevant to chromatin structure. For the multi-track scenarios, we incrementally added epigenetic tracks as follows: 2 tracks (CTCF + H3K4me3), 4 tracks (CTCF + H3K4me3 + H3K9me3 + H3K27ac), and 6 tracks (CTCF + H3K4me3 + H3K9me3 + H3K27ac + H3K27me3 + H3K36me3). These experiments demonstrate that even with a single track, our model achieves robust performance with only a slight reduction in the Spearman correlation coefficient. Incorporating additional tracks consistently enhances performance, showing the advantage of leveraging multiple epigenetic signals (see updated Figure 5c and its legend below).

c Enhanced Predictive Performance with Additional Tracks: A bar graph comparing the Spearman correlation coefficient of EpiVerse’s predictive accuracy for Capan-1 and PANC-1 datasets. The graph highlights improvements as additional epigenetic tracks are integrated: 1 track (CTCF), 2 tracks (CTCF + H3K4me3), 4 tracks (CTCF + H3K4me3 + H3K9me3 + H3K27ac), and 6 tracks (CTCF + H3K4me3 + H3K9me3 + H3K27ac + H3K27me3 + H3K36me3).

To further clarify EpiVerse’s minimum input requirements and their impact, we also added the following statement in the corresponding paragraph:

“Further validation across chromosome 15 revealed that while one track is sufficient for Hi-C contact map prediction (as per the original Avocado paper), incorporating more experimental tracks improves accuracy (Figure 5c).”

We also made this requirement more explicit in the Introduction:

“Unlike traditional epigenetic approaches, the virtual epigenome encompasses computationally imputed or reconstructed epigenetic data constructed through advanced machine learning techniques that integrate diverse datasets, including but not limited to, histone modification patterns, transcription factor binding profiles, RNA expression, and DNA accessibility data, and can be constructed with as few as one ChIP-seq track.”

- 2) The performance of C.Origami is surprisingly poor – significantly worse than that of Orca – which is contradictory to the findings reported in the original paper (DOI: 10.1038/s41587-022-01612-8). It is important to note that the authors re-trained the C.Origami models and compared their EpiVerse models to these re-trained C.Origami models. While the original C.Origami models were trained using DNA sequences, CTCF ChIP-Seq, and ATAC-Seq data, the authors replaced the ATAC-Seq data with DNase-Seq data during the model re-training. Although ATAC-Seq

and DNase-Seq are generally similar in profiling chromatin accessible regions, this substitution could potentially explain the unexpectedly poor performance of C.Origami in their evaluation. To ensure a fair comparison, the authors should re-evaluate the performance of C.Origami using the officially available models and data from <https://zenodo.org/records/7226561>.

We sincerely appreciate the reviewer's insightful feedback. We used DNase-seq instead of ATAC-seq to ensure a fair comparison by maintaining consistent input (Avocado epigenetic signals) and target (Hi-C ground truth) data across all models (except for Orca, which requires only DNA sequence as input). For HiCReg and C.Origami, where re-training was feasible, we re-trained them using the same dataset configurations to allow direct performance comparison under identical conditions. However, we agree with the reviewer that using the officially released pre-trained model could lead to better performance and a more realistic comparison.

We have now re-evaluated C.Origami using its pre-trained models and data from the official repository. We updated Figure 2 with the results from these pre-trained models. With the pre-trained model, C.Origami consistently outperforms Orca in cross-cell type predictions across five evaluation metrics, which aligns with the results reported in the original C.Origami paper.

We have revised the Methods section under "Evaluation of EpiVerse and other state-of-the-art models" to clarify our evaluation process. Specifically, it now states:

“For C.Origami, we utilized the pre-trained model weights and data available from C.Origami’s publicly accessible Zenodo repository.”

We also removed the statements referring to the performance metrics of the old models we trained, which appeared multiple times in the previous manuscript.

- 3) Additionally, upon closer examination of the Methods section, I noticed that specific post-processing steps were applied to different methods to ensure they were compared at the same resolution. However, these processes, including matrix resizing based on skimage's resize function, might have altered the value distributions of other methods, leading to an unfair comparison. What are the results of the same metrics for each method before these post-processing steps?

We appreciate the reviewer's concern. The post-processing step of resizing the matrix was necessary due to differences in the resolution output of different models. The metrics calculated without normalizing the dimensions were not comparable between methods. Without resizing, direct comparisons were not feasible. In fact, the post-processing step followed the same C.Origami methodology for resizing, as outlined in their public source code. We believe the main reason for the suboptimal performance of C.Origami has been addressed by using the pre-trained model, as mentioned in our response to the previous comment. Therefore, we decided to retain the same post-processing step for comparison. We have also added a description explicitly stating the skimage resize function used in the Methods section under "Evaluation of EpiVerse and other state-of-the-art models." The following description was updated:

“On the other hand, C.Origami's output, a 256x256 matrix representing a 2097132 x 2097132bp region, necessitated further processing. We sliced this matrix to 244x244 to approximate a 2M region and applied an exponential transform to revert the original log-transformed Hi-C values. Subsequently, the matrix was resized to 400x400 resolution (equivalent to 5kb/bin) using skimage's resize function, followed by the distance-stratified normalization method proposed by C.Origami, which normalizes each diagonal based on its mean and standard deviation to align with the ground truth. We then focused on the central 1M area (200x200 matrix).”

- 4) The pretrained Avocado model used by this work can impute epigenetic signals for 84 assays (<https://zenodo.org/records/4774521>); however, the authors selected only 71 of these for training the EpiVerse models. They should at least provide an explanation for why these 71 tracks were chosen and why the others were excluded.

We apologize for the oversight. We used only 71 out of 84 tracks simply because these 71 tracks were deemed high-quality by ENCODE Consortium, as described

in the ENCODE3 paper *Completing the ENCODE3 compendium yields accurate imputations across a variety of assays and human biosamples*, detailed in its Additional file 2. To clarify, we have updated the Results section under "EpiVerse robustly imputes cell type-specific chromatin structure" as follows:

*"Leveraging the ENCODE Consortium's existing imputations by Avocado, we retrieved comprehensive epigenetic datasets, including 71 tracks **deemed high quality by the consortium** (Supplementary Table 1)."*

- 5) Probably due to the huge list of dependent packages (<https://github.com/jhhung/EpiVerse/blob/main/environments/HiConformer.yml>), I was unable to set up the HiConformer environment successfully due to frequent conflicts. As a methods paper, it is essential for the authors to simplify the installation procedure and thoroughly test their code on multiple machines to ensure it is runnable.

We apologize for the inconvenience. We have now updated the public GitHub repository (<https://github.com/jhhung/EpiVerse>), including the README.md and related code, to simplify the installation process and address any potential conflicts.

Other Issues

- 1) On page 9, the authors state that "... we retrieved comprehensive epigenetic datasets, which included 71 selected tracks of epigenetic signals (Table S1). These tracks covered a broad spectrum of epigenetic markers, including histone modifications, transcription factor (TF) binding sites, RNA expression profiles, and ATAC-seq." However, "ATAC-Seq" is missing from Table S1.

We sincerely apologize for the misleading information. We have updated the Results section under "EpiVerse robustly imputes cell type-specific chromatin structure" as follows:

*"These tracks covered a broad spectrum of epigenetic markers, including histone modifications, transcription factor (TF) binding sites, RNA expression profiles, and **DNase-seq.**"*

We have also updated the Supplementary Table 1 for further clarification.

Supplementary Table 1: 71 selected (high-quality) tracks of epigenetic signals in Avocado.

Feature Type	Feature
ChIP-seq	ATF3, BHLHE40, CEBPB, CHD2, CTCF, EGR1, ELF1, ELK1, ETS1, EZH2, EZH2phosphoT487, FOXA1, FOXK2, GABPA, GTF2F1, H2AFZ, H2AK5ac, H2BK120ac, H2BK12ac, H2BK15ac, H2BK5ac, H3F3A, H3K14ac, H3K18ac, H3K23ac, H3K27ac, H3K27me3, H3K36me3, H3K4ac, H3K4me1, H3K4me2, H3K4me3, H3K79me1, H3K79me2, H3K9ac, H3K9me3, H4K20me1, H4K8ac, H4K91ac, HDAC2, JUND, KDM1A, MAFK, MAX, MAZ, NRF1, POLR2A, POLR2AphosphoS5, RAD21, RCOR1, REST, RFX5, RXRA, SIN3A, SMC3, SP1, SUZ12, TAF1, TARDBP, TBP, TCF12, USF1, USF2, YY1, ZBTB33, ZFP36
DNase-seq	DNase
polyA_RNA-seq	polyA_RNA-seq_minus, polyA_RNA-seq_plus
total_RNA-seq	total_RNA-seq_minus, total_RNA-seq_plus

- 2) The precise definition of SSIM, an important metric used for model evaluation in this study, is not provided.

We sincerely apologize for the lack of a clear definition of SSIM. We have added a new section titled Evaluation Metrics in the Methods section, which now includes a detailed definition of SSIM.

“Structural Similarity Index (SSIM) evaluates the structural similarity between two Hi-C matrices.

$$SSIM(X, Y) = \frac{(2\mu_X\mu_Y + C_1)(2\sigma_{XY} + C_2)}{(\mu_X^2 + \mu_Y^2 + C_1)(\sigma_X^2 + \sigma_Y^2 + C_2)}$$

where μ_X and μ_Y are the mean values of the predicted and ground truth matrices. σ_X^2 and σ_Y^2 are their variances. σ_{XY} is the covariance between the two matrices. C_1 and C_2 are constants to avoid division by zero.”

- 3) What are the values shown in Figure 2d? Are they average correlations between the predicted contact signals and the experimental signals? If so, are they Pearson correlations, Spearman correlations, or distance-stratified correlations?

Thank you for bringing this to our attention. The metric shown in Figure 2d is the SSIM. We have clarified this in the figure's legend:

“d. Cross-cell type prediction validation: The table shows the SSIM performance for EpiVerse, HiC-Reg, C.Origami, and Orca across different cell types (IMR90, GM12878, K562), with EpiVerse consistently delivering high-quality predictions, demonstrating robust cross-cell-type generalization.”

- 4) In the first paragraph on page 13, the authors state, “Interestingly, we observed that the motifs identified by the sequence encoder CNN are not present in the imputed epigenetic signals.” However, no figures are provided to support this statement.

We apologize for the omission and the associated error. We reached our conclusion by matching the TF names with the top 5 significant motifs identified by each filter using TOMTOM. This updated list is now available in Supplementary Table 3. However, upon closer examination, our name-matching process omitted variations of the same TF, such as **MAFK_DBD** and **RXRA::VDR**, resulting in five TFs that should have been counted. Although this mistake occurred, we believe it still supports the idea that the sequence encoder compensates for features absent in the virtual epigenomes. Consequently, we have updated the statement to address this issue as follows:

“Interestingly, we observed that *most* motifs identified by the sequence encoder CNN are not present in the imputed epigenetic signals. This suggests that the features extracted by the sequence encoder are potentially compensating for the features that are not provided by the virtual epigenomes (Supplementary Table 3).”

Supplementary Table 2 HiConformer Sequence Encoder Motif Matching by TOMTOM. This table shows the top five matching motifs for each filter in the HiConformer sequence encoder

Filter id	Top 5 Matched Motifs
Filter 0	ZNF384, Irx6_2623.2, Mtf1_secondary, ONECUT1, Irx5_2385.1
Filter 1	ONECUT1, Nkx3-2, HOXB13, SOX7_full_2, SOX9_full_4
Filter 2	MAFF, Mafk_primary, MAFK_full_1, MAFK_DBD_1, MZF1
Filter 3	Irx5_2385.1, Irx4_2242.3, Irx6_2623.2, Irx2_0900.3, Irx3_2226.1
Filter 4	POU2F3, Stat5a, Mafb_secondary, PHOX2B, POU3F3
Filter 5	Mafb, Mafb_DBD_1, Mafk_primary, MAFF, SPI1_full
Filter 6	E2F2_secondary, E2F3_secondary, ZNF257, ZNF454,
Filter 7	Mtf1_secondary, ONECUT1, PHOX2B, Foxa2_primary, ZNF384
Filter 8	Mafk_primary, POU2F3, NRL_DBD, ZNF317, MAFF
Filter 9	Irx6_2623.2, Irx5_2385.1, Irx3_2226.1, Irx4_2242.3, Irx2_0900.3
Filter 10	ONECUT1, Irx5_2385.1, Irx4_2242.3, Sox5_primary, Sox13_primary
Filter 11	Pou3f3_3235.2, ONECUT1, POU2F3, Mafb_secondary, Foxl1_primary
Filter 12	ONECUT1, Hoxd13_2356.1, OTX2, Hnf4a_secondary, Irx5_2385.1
Filter 13	PHOX2B, Tbp_primary, MEF2A, Irx5_2385.1, GATA6
Filter 14	SOX2, Tcf3_primary, ONECUT1, Irx6_2623.2, Irx5_2385.1
Filter 15	Irx6_2623.2, Irx5_2385.1, Irx2_0900.3, Irx3_2226.1, Foxg1_DBD_1
Filter 16	CDX2, Sox5_primary, Mtf1_secondary, ONECUT1, PHOX2B
Filter 17	Sox21_primary, Tlx2_3498.2, Zfp105_primary, Tbp_primary, SATB1
Filter 18	POU2F3, Mtf1_secondary, MSX1_DBD_1, MSX2_DBD_1, Msx3_DBD_1
Filter 19	Irx5_2385.1, ONECUT1, Irx4_2242.3, HOMEZ_DBD, FOXA1
Filter 20	Nkx3-2, IRC900814_primary, FOXB1_DBD_2, Irx5_2385.1, HOMEZ_DBD
Filter 21	TBX19, TBX19_DBD, Irx2_0900.3, RORA_DBD_1,
Filter 22	ONECUT1, POU2F3, PHOX2B, Mtf1_secondary, Irx6_2623.2

Filter 23	Nkx3-2, PHOX2B, Sox17_primary, Irx6_2623.2, ONECUT1
Filter 24	Irx5_2385.1, Irx4_2242.3, Irx3_2226.1, Irx6_2623.2, Irx2_0900.3
Filter 25	CEBPA, CEBPD, Mafb_primary, PHOX2B, Mafb_secondary
Filter 26	ONECUT1, Mafk_primary, MAFF, Pax7, Lhx3_3431.1
Filter 27	ONECUT1, Irx6_2623.2, Dbx1_3486.1, Irx5_2385.1, PHOX2B
Filter 28	Irx5_2385.1, Irx3_2226.1, Irx3_0920.1, Irx4_2242.3, Irx2_0900.3
Filter 29	MZF1, Sox13_secondary, Osr2_secondary, Egr1_mouse_DBD_mutant_DBD, TBX4_DBD_2
Filter 30	ONECUT1, Nr2e1, NR2E1_full_1, Nr2e1_DBD_1, ONECUT3
Filter 31	Irx5_2385.1, PHOX2B, Irx4_2242.3, Irx3_2226.1, Tbp_primary
Filter 32	Sox5_primary, Sox13_primary, Foxj1_primary, Sox12_primary, Sox8_primary)
Filter 33	Sox13_secondary, OTX2, PRDM1, Gata3_primary, Pou3f3_3235.2
Filter 34	ZNF211, Hoxc10_2779.2, Zfp410_primary, Osr2_secondary, Hoxa11_2218.1
Filter 35	Isgf3g_primary, Mtf1_secondary, ZNF384, IRC900814_primary, ONECUT1
Filter 36	Elf5, EEWSR1-FLI1, ZNF530, EHF, ETV1
Filter 37	Mtf1_secondary, GATA6, Irx5_2385.1, ONECUT1, Irx4_2242.3
Filter 38	FOXB1_DBD_2, GATA6, MEF2C, MZF1, NFIX_full_3
Filter 39	ONECUT1, Tbp_primary, ZNF384, Foxj1_primary, PHOX2B
Filter 40	Elf5, Nfatc2, Obox1_3970.2, Nfat5, Nfatc1
Filter 41	Nkx3-2, Nkx2-9_3082.1, Nkx3-1_2923.2, POU3F3, POU3F3_DBD_1
Filter 42	ONECUT1, Irx5_2385.1, CDX2, POU2F3, Mafb_secondary
Filter 43	HOMEZ_DBD, Homez_1063.2, Sox21_primary, Pax7, Sox13_primary
Filter 44	ZNF384, ONECUT1, Srf_secondary, Zfp105_primary, Mtf1_secondary
Filter 45	Mtf1_secondary, ONECUT1, Sox5_primary, Sox21_primary, Foxa2_primary
Filter 46	ONECUT1, ONECUT3, ONECUT3_DBD, Mtf1_secondary, GATA6
Filter 47	ZNF211, Hoxc10_2779.2, POU2F3, Hoxc11_3718.2, Zfp128_secondary
Filter 48	CEBPA, POU2F3, Pou3f3_3235.2, EOMES, EOMES_DBD_1
Filter 49	Srf_secondary, HOMEZ_DBD, Nkx3-2, Homez_1063.2, Sox13_primary
Filter 50	PHOX2B, SOX2, OTX2, Sox15_secondary, Nfatc2
Filter 51	Stat5a, MEF2C, Sox21_primary, PHOX2B, NFIX_full_4
Filter 52	Homez_1063.2, RUNX3, Nkx3-2, GRHL1, GRHL1_full
Filter 53	Nfat5, ZNF354A, Nfatc1, Pax7, SOX13

Filter 54	ZNF384, Mafk_secondary, SATB1, Irx5_2385.1,
Filter 55	RXRA::VDR, Lef1_primary, LEF1_DBD, ZFP14, Vdr_DBD
Filter 56	RUNX3, IRC900814_secondary, Irx5_2385.1, Irx6_2623.2, Irx3_2226.1
Filter 57	Sox17_primary, ONECUT1, SOX9_full_5, Foxj3_secondary, Nkx3-2
Filter 58	PRDM1, SPIC, SPIC_full, Spic_DBD, ZNF418
Filter 59	Srf_primary, POU3F3, POU3F3_DBD_1, NFATC1_full_1, POU2F3
Filter 60	Nkx3-2, ZIM3, JUNB, TBX21_full_3, BATF3
Filter 61	Irx5_2385.1, Irx4_2242.3, Pax7, Irx3_2226.1, Irx6_2623.2
Filter 62	Mtfl_secondary, GATA6, ONECUT1, Sox21_primary, ZNF384
Filter 63	Irx6_2623.2, Irx5_2385.1, Nfat5, Irx3_2226.1, Irx2_0900.3

- 5) Genomic coordinates should be added to all figures containing contact heatmaps to facilitate reproducibility of the analysis.

We appreciate the reviewer's valuable suggestion. We have now added genomic coordinates to all figures containing contact heatmaps, including Figure 2a, Figure 3d, Figure 5b, and the Supplementary Figures to facilitate reproducibility of the analysis.

- 6) In Figure 3d, the loop shown in the center appears to be a canonical CTCF loop. It would be more informative to overlay both enhancer signals and CTCF signals on top of the Hi-C heatmaps. Additionally, there seems to be a typo in the corresponding figure caption. Based on the context, it should be the blue color in the rightmost column that indicates regions with stronger interactions in K562, rather than the red color.

Thank you for your valuable suggestion. We have now overlaid the enhancer signals and CTCF signals on top of the Hi-C contact map in Figure 3d. The figure illustrates that K562 has more active enhancers than GM12878, potentially highlighting more tissue-specific regions.

Additionally, we have corrected the typo in the figure's legend:

“red indicates regions with stronger interactions in GM12878, while blue denotes stronger interactions in K562.”

- 7) In Figure S9, the caption states that “Each panel presents Hi-C contact maps for selected genomic regions, showcasing the distribution of active enhancers ...” However, only heatmaps are shown for each region, and the enhancers are not displayed.

We appreciate the reviewer's suggestion. We have updated Figure S9 (now Supplementary Figure 10) by adding active enhancer signals below each Hi-C contact map. The figure now clearly shows that, from left to right, the number of enhancers decreases, and correspondingly, the Hi-C contact intensity also decreases.

- 8) The figure caption for Figure 4d does not align with the text that references this figure. According to the caption, this figure shows the enrichment of gene sets associated with active enhancers. However, the text suggests that this figure should demonstrate GO terms of genes in regions with the largest variation in chromatin structure.

We sincerely apologize for the typographical error. We have corrected the legend for Figure 4d as follows:

“d. GO terms of Genes in Hi-C Largest Variation Regions: GO term analysis of genes located in regions with the largest variation in chromatin structure highlights strong tissue specificity. This figure demonstrates the critical role of chromatin organization in defining tissue- and cell-specific functions, with significant enrichment in skin, tonsil, breast, tongue, bronchial epithelial cells, colon, and NHEK.”

- 9) In Figure S12, where are the promoter-based results? Based on the main text that references this figure (Page 18), one would expect two sets of analyses: one using gene promoters and another using active enhancers.

We apologize for not providing clear information. We have revised the Results Section *EpiVerse enables the analysis of whole human tissue chromatin structure* to clarify the analysis:

"To investigate whether both promoter hubness and enhancer activity serve as reliable indicators of tissue-specific gene expression, we selected 19 tissues from the 39 we imputed that are also available in the GTEx Portal. We then employed weighted Kolmogorov-Smirnov testing using GSEA to evaluate the effectiveness of promoter hubness and active enhancers in identifying tissue-specific gene expression profiles. A tissue was considered a "hit" if its gene ranking by Virtual 4C promoter hubness or active enhancers appeared in the top 5 significant results, indicating that the ranked genes successfully identified its own gene expression profile through GSEA. Our analysis found that relying solely on promoter hubness for gene ranking correlated less with tissue-specific expression than expected, achieving only 2 hits out of 19 tissues. Conversely, active enhancers proved to be a superior metric, more accurately ranking genes and serving as a more reliable indicator for characterizing tissue-specific gene expression patterns, as demonstrated in Supplementary Figure 13 (12 hits out of 19 tissues)."

We have also updated the Supplementary Figure 13 to match the main text.

a**b**
Supplementary Figure 1 Comparative Analysis of Tissue-Specific Gene Identification by Promoters and Active Enhancers Using Gene Set Enrichment Analysis (GSEA)³.

a. Active enhancer-based GSEA plots demonstrate tissue-specific gene expression patterns identified across various tissues. The criterion for an active enhancer's effectiveness in pinpointing tissue-specific gene expression is its inclusion in the top 5 GSEA ranks for the target tissue. The analysis highlights that active enhancers are successful in identifying the correct tissue-specific gene expression profiles in 12 out of the 19 tissues when this criterion is met, specifically in the adrenal gland, aorta, B cells, brain, colonic mucosa, esophagus muscularis mucosa, esophagus squamous epithelium, heart (left ventricle and right ventricle), natural killer cells, psoas muscle, and the right lobe of liver.

b. Promoter-based GSEA plots display tissue-specific gene expression identification across the same set of tissues with active enhancer-based GSEA plots. Promoters only successfully identify tissue-specific gene expression profiles in the brain and right lobe of the liver.

Labels such as "70-79 Up" refer to the age range of individuals in the GTEx dataset (e.g., 70-79 years), with "Up" indicating upregulated genes in that age group compared to baseline.

10) In addition to the missing results in Figure S12, what do the labels such as "70-79 Up" and "40-49 Up" mean in the figure legend?

We apologize for the unclear information again. We have clarified the legend for Figure S12 (now Supplementary Figure 13):

"Labels such as "70-79 Up" refer to the age range of individuals in the GTEx dataset (e.g., 70-79 years), with "Up" indicating upregulated genes in that age group compared to baseline."

11) In the main text that references Figure S15, the authors state, "This in silico perturbational Hi-C experiments by EpiVerse mirrored the results of the original study, yielding new chromatin contacts and reinforced domain boundaries, as evidenced by the congruence with real Hi-C experiment observations." Based on this description, one would expect Figure S15 to include both predicted and observed contact matrices, demonstrating their agreement. However, the real observed contact matrices are missing from this figure.

We appreciate the reviewer's observation regarding Figure S15 (now Supplementary Figure 16). It is worth noting that the original study was conducted in the HAP1 cell line, which is not available in the Avocado pre-trained model. However, since the DNA insertion is known to create a tissue-invariant domain

boundary, we expected to observe similar effects in other tissues. Therefore, we used the imputed H1 cell line data (as used in the Orca paper) to demonstrate comparable boundary-forming effects in our predictions. Despite the differences in cell lines, we successfully replicated key effects observed in the original study, such as de novo contact formation and additive boundary strengthening.

Our purpose was to highlight the new chromatin contacts at the insertion site, and we originally thought including the ground truth for the 1M region could potentially confuse readers due to other irrelevant HAP1- or H1-specific interaction changes. However, we agree with the reviewer that the real observed contact matrices are still a valuable reference. To address this, we have updated Figure S15 (now Supplementary Figure 16) to include both the predicted and observed contact matrices, providing a clearer comparison to demonstrate their agreement. This ensures that the congruence between the predicted and real Hi-C contact maps is visually evident, effectively supporting our claims. The figure legend has also been updated to clarify the cell line differences between the ground truth (HAP1) and the prediction (H1), ensuring readers understand the rationale behind using H1 data in place of HAP1.

Supplementary Figure 2 in silico insertion Hi-C experiment². The figure shows both the predicted and observed contact matrices for the DNA insertion experiment. The original study was conducted in the HAP1 cell line, which is unavailable in the Avocado pre-trained model. Therefore, imputed H1 cell line data was used to demonstrate similar boundary-forming effects in our predictions. Despite the differences in cell lines, the comparison illustrates the congruence between EpiVerse's in silico perturbations and real Hi-C data, showing new chromatin contacts and reinforced domain boundaries. Arrows indicate insertion sites. **a. De novo Contact Formation:** This figure illustrates the creation of new chromatin contacts following insertion of a 2-kb DNA sequence at a tissue-invariant domain boundary. The heatmap comparison clearly shows the emergence of interactions that were not present prior to the insertion event. **b. Additive Boundary Strengthening:** This figure depicts the reinforcement of existing chromatin boundaries post-insertion. The heatmaps contrast the chromatin interaction patterns before and after the insertion, highlighting the enhanced boundary definition as a result of the experiment.

12) In Figure S16, the column labels are missing. Additionally, to provide a more comprehensive view of the locus and the associated dynamics, genes, RNA-Seq data, and relevant epigenetic signals should be overlaid on top of the heatmaps.

We appreciate your insightful comments regarding the *in silico* Hi-C deletion experiment. We agree that showing those tracks could be helpful. We have added CTCF, H3K27ac, total RNA-seq (minus and plus strands), and important gene information to the figures to provide a more comprehensive view with the original Hi-C matrices. For Figure S16 (now Supplementary Figure 17), after the deletion of

the TAL1 and LMO2 neighborhood boundary, we can visualize an elevation in the Hi-C contact frequencies around these oncogenes, leading to their activation. Please note that since Avocado does not support imputation for missing signals (e.g., DNA deletions), and EpiVerse predicts the contact map by zeroing the input corresponding to the deleted region, no track is shown following the deletion.

Supplementary Figure 3 *In silico* deletion Hi-C experiment³.

a. TAL1 Oncogene Activation: This figure presents the consequences of deletion near the TAL1 insulated boundary. The heatmaps compare chromatin structures pre- and post-deletion, with arrows highlighting the activation of the TAL1 oncogene due to the deletion event. **b. LMO2 Oncogene Activation:** The figure demonstrates the impact of a targeted deletion near the LMO2 insulated boundary, showing the resultant activation of the LMO2 oncogene. The heatmaps before and after deletion illustrate the changes in chromatin interaction patterns, with arrows indicating the specific areas of alteration.

- 13) There are several issues with figure 5 alone: First, it appears that panels b and c are swapped. Second, in panel b, the epigenetic signals before and after perturbation should be displayed to help readers better understand the relationship between the dynamics of epigenetic signals and chromatin structure. Third, the caption for panel e is missing. Finally, the relevance of panels d and e is unclear, as they do not seem to be related to chromatin interactions.

We sincerely apologize for the incorrect labels and appreciate the reviewer's thoughtful comments. We have corrected the labels and made the following updates:

For Figure 5b, we have added two tracks of histone modifications that showed changes in the original study (H3K4me3 and H3K27ac), along with one track from our Avocado imputations (SMC3). These additional epigenetic signals help illustrate the dynamic relationship between the epigenome and chromatin structure.

For Figures 5d and 5e, our intention was to showcase EpiVerse's ability to use imputed ChromHMM data to identify tissue-specific upregulated genes. This feature allows users to efficiently navigate potential tissue-specific regions, providing valuable insights for further exploration.

We have also refined the figure legends to ensure they align with the content and provide clearer explanations. Please see below.

“Figure 5 In silico perturbational pancreatic cancer metastasis Hi-C experiment a EpiVerse Pipeline for PANC Experiment Reproduction: Schematic diagram illustrating the EpiVerse process from constructing virtual epigenetic signals to chromatin 3D structure modeling, using various histone modifications and CTCF as inputs to accurately replicate chromatin structures. b Comparison between Imputation Hi-C and experimental Hi-C: Heatmaps indicating the Hi-C interaction matrices with substantial variations in H3K27ac and enhancer presence across chromosome 15. The top row presents imputed data, while the bottom row displays experimental Hi-C data for Capan-1, PANC, and their differential heatmap. Gene Discovery via Active Enhancer Counts: Line graph demonstrating the discovery rate of genes using the active enhancer count approach, validating its effectiveness in identifying genes with differential expression. c Enhanced Predictive Performance with Additional Tracks: A performance comparison bar graph showing the improvement in EpiVerse’s predictive accuracy as additional experimental tracks are integrated. d EpiVerse Utilizes Active Enhancers Marking Upregulated Genes in Capan-1: This figure shows the gene identification rate for the top k enhancer-enriched regions associated with upregulated genes in the Capan-1 cell line. The performance of EpiVerse (light blue) is compared to a random model (red). EpiVerse consistently

identifies a higher proportion of relevant genes as more enhancer regions are considered, outperforming the random baseline. e GSEA of Differential Genes in PANC and Capan-1: Analysis of GO terms related to cell motility in differential genes between Capan-1 and PANC-1 cells, highlighting their association with mechanisms of cancer metastasis. “

- 14) On page 20, there is a lack of a literature reference for the sentence: “For the insertion experiment, we were inspired by a pivotal study that highlighted the importance of engineering chromosomal contact domains.”

We sincerely apologize for the missing reference. We have now added the correct citation (Zhang et al.) after the sentence.

- 15) To ensure reproducibility, the pre-trained models for GM12878 and K562 should also be uploaded. Currently, only the model trained in IMR90 has been made available (<https://zenodo.org/records/10889119>).

Thank you for your suggestion. We have uploaded the pre-trained models for GM12878 and K562 to Zenodo, and they are now available at <https://zenodo.org/records/13759557>.

- 16) In the EpiVerse browser (<https://epiverse.jhhlab.tw/>), entering values for “Chromosome,” “Start Location,” and “Tissue Sample,” as shown in Supplementary Figure 13, resulted in an error message: “Error fetching matrices: Please check your input range.” Please verify whether the browser is functioning correctly.

Thank you for taking the time to check our browser. We have fixed the issue and redeployed the website at <https://epiverse.jhhlab.tw/>.

Reviewer #1 (Remarks on code availability):

Probably due to the huge list of dependent packages (<https://github.com/jhhung/EpiVerse/blob/main/environments/HiConformer.yml>), I was unable to set up the HiConformer environment successfully due to frequent conflicts. As a methods paper, it is essential for the authors to simplify the installation procedure and thoroughly test their code on multiple machines to ensure it is runnable.

We sincerely apologize for the oversight regarding the difficulties in setting up the environment. We have now updated the public GitHub repository (<https://github.com/jhhung/EpiVerse>), including the README.md and related code, to

simplify the installation process and address any potential conflicts.

Reviewer #2 (Remarks to the Author):

1. Some content in the Introduction overlaps with the Conclusion and Rationale sections. It's recommended to condense these parts for clarity.

We appreciate the reviewer's suggestion. Our original intent was to provide readers with a clear understanding of the innovative aspects of our methods and the challenges we faced by reiterating them in both the Introduction and Rationale sections. However, we have now condensed the Rationale section to Introduction section and updated the overlapping content in both sections for improved clarity.

2. Predicting chromatin state is a major functionality of EpiVerse. It would be beneficial to expand on this feature and its results. Providing downloadable chromatin states for various cells would offer readers an alternative to chromHMM.

We appreciate your insightful comments. The functionality of predicting ChromHMM is indeed a key feature of EpiVerse, providing biologically meaningful annotations. We have made ChromHMM state visualizations for various tissues available in the EpiVerse Browser (<https://epiverse.jhlab.tw/>), and we have also uploaded our ChromHMM imputations to <https://zenodo.org/records/13759557>.

3. In Fig. 1, the HiConformer model, a core component of this paper, is not described clearly enough. Adding some illustration would help readers better understand the model by combining textual description with a visual representation.

We appreciate the reviewer's suggestion. HiConformer is a key component of EpiVerse, and its Diagonal Extraction Algorithm introduces a novel approach compared to traditional methods. Unlike conventional approaches (Bin-to-Bin), where each Hi-C contact is predicted independently using only features from the specific contact point, the Diagonal Extraction Algorithm allows the model to leverage information from neighboring regions (Area-to-Area). By extracting entire diagonals from the ground truth Hi-C matrices, this method enables each prediction to consider surrounding context, expanding the model's receptive field and improving predictive performance. To clarify this, we have updated the Methods section *Diagonal Extraction Algorithm* and added Supplementary Figure 18 to visually illustrate this concept.

“The Diagonal Extraction Algorithm is designed for expanding model’s receptive field in the chromatin structure prediction task. Unlike traditional approaches (Bin-

to-Bin), where each Hi-C contact is predicted using only the features at that specific contact point, this method leverages information from surrounding regions (Area-to-Area) (Supplementary Figure 18).”

Additionally, we have updated Figure 2c with input data dimensions to further enhance the readers' understanding of the model's overall data flow and innovative aspects.

- All heatmaps (e.g., Fig. 2a and Fig. 2d) should include a color scale legend.

We thank the reviewer for this helpful suggestion. We have added a color scale legend to all heatmaps, including Figure 2a, Figure 2d, Figure 3d, Figure 5b, and the Supplementary Figures.

- All boxplots (e.g., Fig. 4b and Fig. S10) should include a test for statistical significance.

We have included a test for statistical significance in Figure 4b and Figure S10 (now Supplementary Figure 11). We have also updated the figure legends as follows:

“Figure 4b. Boxplot of Hi-C Value Quantile vs. Active Enhancer Counts: This boxplot compares the distribution of Hi-C interaction values across four quantiles to the counts of active enhancers, highlighting a trend that suggests a relationship

*between chromatin interaction frequency and enhancer activity. The two-sided Mann-Whitney test was conducted, with significance indicated by asterisks: * $p < 0.05$, * $p < 0.01$, * $p < 0.001$, and * $p < 0.0001$.*”

*“Supplementary Figure 11. This boxplot compares the distribution of Hi-C interaction values across four quantiles to gene counts, highlighting a trend that suggests a relationship between chromatin interaction frequency and gene counts. The two-sided Mann-Whitney test was conducted, with significance indicated by asterisks: * $p < 0.05$, * $p < 0.01$, * $p < 0.001$, and * $p < 0.0001$.*”

6. Some computational results in this paper can be verified by other literature (e.g., at the end of Page 18, Paragraph 2: This observation is...in transcription levels [33]). Clearly indicate the specific reference position or figure.

We appreciate the reviewer’s suggestion. We have clarified the specific reference positions and figures below for the reviewer to ensure easier traceability for verification.

There are three such statements:

1. In the Results section “EpiVerse discerns elements that characterize chromatin structure by integrating virtual epigenome and chromatin state annotation”, we demonstrated that BMI1 is overexpressed in various types of leukemia, which is related to the signature BMI1_DN_MEL18_DN.V1_DN. BMI1’s overexpression is associated with increased cell proliferation and inhibition of apoptosis and differentiation, hallmark traits of cancer cells [28].

In the abstract of the original study [28], it is stated:

“It has been demonstrated that over-expression of Bmi-1 occurs in a variety of cancers, including several types of leukemia. This gene plays a key role in the self-renewal of stem cells. Leukemic cells lacking Bmi-1 underwent proliferation arrest and showed signs of differentiation and apoptosis. These findings led to the proposal of Bmi-1 as a potential target for therapeutic intervention in cancer.”

2. Additionally, in the Results section “EpiVerse discerns elements that characterize chromatin structure by integrating virtual epigenome and chromatin state annotation”, we found that the PDGF_ERK_DN.v1_DN signature is notable, as PDGF A and PDGF B are genes induced during the megakaryoblastic differentiation of K562 cells [29].

This is corroborated in the original study [29] in the section “*The A- and B-chains of PDGF are induced during megakaryoblastoid differentiation of K562 cells*”, which states:

“During induced differentiation of K562 cells, the PDGF A- and B-chain genes are differentially expressed depending on the induced phenotype of the cells.”

3. In the Results section “EpiVerse enables the analysis of whole human tissue chromatin structure”, we demonstrated: “This observation is in line with recent studies showing that numerous genes can undergo significant changes in 3D connectivity without corresponding changes in transcription levels [37].”

In the original study’s Figure 4b [37], it was shown that “*The genes identified as 3D-insensitive, meaning they show significant changes in 3D chromatin connectivity without corresponding transcriptional changes, are enriched for housekeeping functions such as RNA processing, metabolism, and cell cycle regulation. These genes display high levels of promoter H3K27ac and ATAC-seq signals, suggesting their promoters are constitutively active and less reliant on distal 3D chromatin interactions for transcription.*”

7. In Table S1, please list the specific download addresses for each data source.

We appreciate the reviewer’s comments. Since we imputed many tissues and the data links are lengthy and messy across tracks and tissues, we have provided all the download information for the tracks in the EpiVerse GitHub repository (https://github.com/jhhung/EpiVerse/blob/main/Avocado/Avocado_metadata.csv) instead of Table S1. Additionally, we have provided a crawler script to facilitate the retrieval of the data for each tissue.

8. Each table should include explanation of the terms' name.

We thank the reviewer for the suggestion. Unfortunately, the terms (such as "Module 55" in Supplementary Table 4) defined by MSigDB are somewhat arbitrary and can be confusing without detailed explanations. For example, "Genes in the cancer module 55" from MSigDB only implies an association with certain cancers, making it challenging to provide detailed explanations for each term. To address this, we have added hyperlinks to the corresponding MSigDB entries for each term in the Supplementary Table 4-7, allowing readers to quickly access full descriptions.

- Please ensure that figures associated with chromosomes or genomic regions are marked or annotated with the relevant chromosome or genomic coordinates, such as the 2D heatmaps in Fig. 2a.

We appreciate the reviewer's suggestion. We have added genomic coordinates to all contact maps, including Figure 2a, Figure 3d, Figure 5b, and the Supplementary Figures.

- It is recommended to cite original articles in your references. For instance, the citation for Hi-C technology on the first paragraph on page three should reference its original source: Eriz, Science, 2009.

We sincerely apologize for the oversight. We have added the original source article for Hi-C, Eriz, Science, 2009, and have rechecked all citations throughout the manuscript.

- On the EpiVerse browser (<https://epiverse.jhlab.tw>), it is suggested to provide guidance and examples on the webpage as an error occurs: "Error fetching matrices: Please check your input range."

Thank you for taking the time to check the browser. We have fixed the issue and redeployed the website at <https://epiverse.jhlab.tw/>.

- Fig. 2b should include a y-axis title.

We thank the reviewer for this suggestion. We have added a y-axis title to Figure 2b. Since the figure includes multiple metrics with values ranging between 0 and 1, we used "Value" as the y-axis title.

13. Bar charts, such as in Fig. S7, should include error bars.

We thank the reviewer for the suggestion. We have added error bars to Figure S7 (now Supplementary Figure 8) as well as Supplementary Figure 1a and 1b.

Supplementary Figure 1a and 1b:

Supplementary Figure 8:

14. Page 9, Paragraph 1, the abbreviation “3DIV” needs an explanation.

We thank the reviewer for the suggestion. 3DIV stands for "3D-genome Interaction Viewer and database." We have updated the first instance of 3DIV in the manuscript within the Results section, “EpiVerse robustly imputes cell type-specific chromatin structure” to:

"For the subsequent stages of the EpiVerse training pipeline, we collected the 3D-genome Interaction Viewer and database (3DIV) distance-stratified normalization Hi-C datasets."

15. Page 11, Paragraph 1, provide the full name and explanation for “O/E correlation.”

We thank the reviewer for the suggestion. We have added the full name for O/E correlation at its first mention in the manuscript, in the Results section EpiVerse robustly imputes cell type-specific chromatin structure:

"EpiVerse outperformed other methods in Pearson correlation coefficient, Spearman correlation coefficient, Observed / Expected (O/E) correlation (Figure 2b)."

To provide a more detailed explanation, we have also appended a new section, Evaluation Metrics, under Methods. The precise definition of O/E correlation is as follows:

Observed/Expected (O/E) Correlation compares observed and expected Hi-C contact frequencies for predicted and ground truth data.

$$OE_{ij} = \frac{C_{ij}}{\overline{C}(d)}$$
$$r_{O/E} = \frac{\sum_{i=1}^n (OE_{i,pred} - \overline{OE}_{pred})(OE_{i,gt} - \overline{OE}_{gt})}{\sqrt{\sum_{i=1}^n (OE_{i,pred} - \overline{OE}_{pred})^2} \sqrt{\sum_{i=1}^n (OE_{i,gt} - \overline{OE}_{gt})^2}}$$

where C_{ij} is the observed contact frequency between loci i and j . $\overline{C}(d)$ is the expected contact frequency at distance d . $OE_{i,pred}$ and $OE_{i,gt}$ are the observed / expected values for the predicted and ground truth i_{th} diagonal. n represents the total number of valid diagonals used in the calculation across the Hi-C matrices.

16. Page 12, Paragraph 3, “IG” needs a citation and explanation.

We appreciate the reviewer’s insightful comment. We have added the citation for Integrated Gradient (IG) in the Results section “EpiVerse discerns elements that characterize chromatin structure by integrating virtual epigenome and chromatin state annotation” at its first mention:

*“The EpiVerse pipeline is applied to four tasks: DNA sequence motif discovery, assessment of the importance of epigenetic signals for chromatin structure using the **Integrated Gradient (IG)**²², analysis of the impact of epigenetic signals on enhancer states, and exploration of the interplay between enhancers and chromatin structure.”*

Additionally, we have expanded the explanation of IG in the same section:

“For the assessment of the importance of epigenetic signals for chromatin structure, we calculate the IG of epigenomic signals from HiConformer (Supplementary Figure 9; see Methods). IG is a technique used to attribute the contribution of individual features (in this case, epigenetic signals) to a model’s prediction.”

17. Page 13, Paragraph 1, “PWM” needs a citation and explanation.

We appreciate the reviewer’s reminder. We have updated the Results section “EpiVerse discerns elements that characterize chromatin structure by integrating virtual epigenome and chromatin state annotation” to include a citation and a brief explanation of PWM:

*“For DNA sequence motif discovery, we analyzed the first-layer CNN activations of the sequence encoder in HiConformer (Supplementary Figure 9; see Methods). **These activations were filtered and processed to construct Position Weight Matrices (PWMs)**²³, which represent the nucleotide frequency at each position within a motif, capturing sequence variability across multiple binding sites. The PWMs derived from the CNN filters correspond to the consensus sequences identified by the individual CNN kernels, effectively capturing recurring patterns or motifs in the sequence data (Figure 3a).”*

18. Page 13, Paragraph 2, "GSEA" needs an explanation.

We appreciate the reviewer’s feedback. We have added the following explanation of GSEA in the Results section “EpiVerse discerns elements that characterize

chromatin structure by integrating virtual epigenome and chromatin state annotation”:

*"By ranking differential gene importance revealed by IG and conducting a preranked **Gene Set Enrichment Analysis (GSEA)** we observed that K562 is enriched in cancer-related gene set, consistent with its nature as a leukemia cancer cell line, as opposed to GM12878, a normal lymphocyte cell line. GSEA is a computational method used to determine whether predefined sets of genes show statistically significant, coordinated differences between two biological conditions, providing insights into underlying biological processes."*

19. Page 13, Paragraph 2, check "v" cases for “BMI1_DN_MEL18_DN.V1_DN. BMI1” and “PDGF_ERK_DN.v1_DN.”

Thank you for pointing this out. The lowercase 'v' in 'BMI1_DN_MEL18_DN.V1_DN. BMI1' and 'PDGF_ERK_DN.v1_DN' was indeed a typo. We have corrected it in the revised manuscript.

20. In Fig. 3c, since the data is sparse and colors are similar, label factor names directly in the figure for easier identification.

We appreciate the reviewer’s helpful suggestion. We have now labeled the factor names directly in the figure, making it more intuitive and easier to interpret.

21. At the end of Fig. 3d's caption, the color description may be reversed.

We sincerely apologize for the oversight. We have corrected the typo in the figure caption as follows:

“Figure 3d. Largest Differential Enhancer Regions: The heatmaps display the chromatin interaction differences, highlighting regions with significantly varied enhancer activity between the two cell lines. Stronger chromatin looping interactions are evident in K562 as compared to GM12878. The top row represents the imputed chromatin interactions, while the bottom row shows the experimental data. The rightmost column presents the differential heatmaps, where red indicates regions with stronger interactions in GM12878 and blue denotes stronger interactions in K562.”

22. Page 29, Paragraph 2, provide information monitoring the HiConformer training process, likewise epoch, loss, AUROC, and other diagnostic curves.

We appreciate the reviewer's valuable suggestion. We have provided detailed information on the HiConformer training process, including the loss curve and various metrics tracked during training on IMR90 in Supplementary Figure 7.

Supplementary Figure 4 EpiVerse Training Convergence and Performance Metrics in IMR90.

a Epoch Loss: Shows overall loss reduction for training and validation. **b Hi-C Loss:** Demonstrates the Hi-C loss convergence during training. **c Bin1 ChromHMM State Loss:** Tracks the loss for Bin1 ChromHMM state prediction. **d Bin2 ChromHMM State Loss:** Tracks the loss for Bin2 ChromHMM state prediction. **e Hi-C Weighted Pearson Correlation:** Correlation between predicted and actual Hi-C data. **f Hi-C Weighted Distance-Stratified Pearson Correlation:** Measures distance-stratified Pearson correlation between predicted and actual Hi-C data. **g Bin1 ChromHMM State F1-Score:** Indicates precision and recall balance for Bin1. **h Bin2 ChromHMM State F1-Score:** Indicates precision and recall balance for Bin2.

These results have been also added to the Results section *EpiVerse robustly imputes cell type-specific chromatin structure*, as they are crucial in demonstrating that our model successfully converged:

"Moreover, EpiVerse exhibited high reproducibility; retraining the model multiple times yielded consistently reliable results (Supplementary Figure 6b). The loss curve further confirms that EpiVerse converges well during training (Supplementary Figure 7). Additionally, in evaluating its capability to predict ChromHMM states, EpiVerse achieved an average accuracy rate of over 90% (Supplementary Figure 8), demonstrating its proficiency in utilizing epigenetic signals and capturing the complexities of chromatin state dynamics."

23. In Fig. 5, panel "b" and "c" reversed with their captions.

We sincerely apologize for the mistake. We have corrected the error.

24. It is more precise to use “by up to 200% improvement” than “by a 200% improvement” in paragraph 2 of the Conclusions section.

We appreciate the reviewer's suggestion. Based on feedback from other reviewers, we re-evaluated the performance using C.Origami pre-trained weights due to the unexpectedly lower performance compared to that reported in the original paper. The updated metrics have been revised, and the previous claim of a 200% improvement is no longer valid. Consequently, we have updated the performance description in the Conclusions section to:

“EpiVerse's ability to predict cell type-specific chromatin structures outperformed existing models across all performance metrics, setting it apart from other models.”

Reviewer #2 (Remarks on code availability):

In the README on EpiVerse's GitHub, there are instructions for installing three software components. I tried to install them on my computer (ProductName: macOS; ProductVersion: 11.4; BuildVersion: 20F71). As a result, the “Avocado Environment” and the “fanc Environment” were successfully installed. However, an error occurred while installing the “HiConformer training and MIRNet training Environment” [truncated for brevity].

We sincerely apologize for the oversight regarding the difficulties in setting up the environment. We have now updated the public GitHub repository

(<https://github.com/jhhung/EpiVerse>), including the README.md and related code, to simplify the installation process and address any potential conflicts.

Reviewer #3 (Remarks to the Author):

1. What is the difference between Figure 2 and Supplementary Figure 1? The authors clearly state what they are trying to show, including stating the cell lines that are being tested, to lessen the confusion instead of copying and pasting the same caption for both Figures. The overall captions for all the figures can be more descriptive.

We appreciate the reviewer's insightful feedback. Initially, we included Supplementary Figure 1 to offer readers a visual representation of an additional genomic region. However, we agree with the reviewer's concern that the figure could lead to potential confusion. To enhance clarity, we have now removed Supplementary Figure 1 from the manuscript.

2. On page 9, a brief description of "hybrid L1-Diagonal-corr loss function" would be beneficial.

We greatly appreciate your suggestion. To better align with the description of the MIRNet loss in the Methods section, we have replaced the term "hybrid L1-Diagonal-corr loss function" with "surrogate loss", and added a brief description of its purpose in the Results section under 'An Overview of EpiVerse's Framework,' to include a more detailed explanation of this loss function, highlighting its novel aspects:

"Lastly, the MIRNet module, which processes diagonals predicted by HiConformer, was trained using a novel surrogate loss function. This loss combines diagonal correlation, which captures distance-stratified Hi-C relationships, and Charbonnier loss for robustness against outliers. The surrogate loss uniquely balances prediction accuracy with structural consistency, improving the quality of interaction data."

3. O/E correlation is explained for the first time on page 33. The authors should introduce this when they first use the term on page 11. I.e. observed / expected (O/E) correlation (Figure 2b).

We thank the reviewer for the reminder. We have now added the full name when Observed / Expected (O/E) correlation is first mentioned:

"EpiVerse outperformed other methods in Pearson correlation coefficient, Spearman correlation coefficient, Observed / Expected (O/E) correlation (Figure

2b)."

Additionally, we have included a detailed definition of O/E correlation in the Methods section under Evaluation Metrics for clarity.

“Observed/Expected (O/E) Correlation compares observed and expected Hi-C contact frequencies for predicted and ground truth data.

$$OE_{ij} = \frac{C_{ij}}{\overline{C}(d)}$$
$$r_{O/E} = \frac{\sum_{i=1}^n (OE_{i,pred} - \overline{OE}_{pred})(OE_{i,gt} - \overline{OE}_{gt})}{\sqrt{\sum_{i=1}^n (OE_{i,pred} - \overline{OE}_{pred})^2} \sqrt{\sum_{i=1}^n (OE_{i,gt} - \overline{OE}_{gt})^2}}$$

where C_{ij} is the observed contact frequency between loci i and j . $\overline{C}(d)$ is the expected contact frequency at distance d . $OE_{i,pred}$ and $OE_{i,gt}$ are the observed / expected values for the predicted and ground truth i_{th} diagonal. n represents the total number of valid diagonals used in the calculation across the Hi-C matrices.”

4. What was the reason behind choosing “chromosomes 1 through 18 were designated for the training set, chromosomes 19 and 20 for the validation set, and chromosomes 21 and 22 were used as the testing set”? Is it due to the decreasing chromosome sizes?

Yes, our goal was to create a generalized model capable of cross-cell type Hi-C prediction. To achieve this, we aimed to maximize the amount of training data, which is why we selected the largest chromosomes for the training set. This allows the model to learn from the most data possible.

5. The authors should highlight that EpiVerse is publicly available on GitHub in the abstract.

We thank the reviewer's comment. While we had previously included the GitHub link in the Code Availability section, we have now also added the GitHub link in the Abstract to make it more prominent.

“Furthermore, EpiVerse facilitates unprecedented in silico perturbational experiments at the “epigenome-level” to unveil the chromatin architecture under

specific conditions. EpiVerse is available on <https://github.com/jhhung/EpiVerse>.”

6. What is the reproducibility of EpiVerse? Showing this will greatly benefit and strengthen the manuscript.

We appreciate the reviewer's comment and understand the importance of model reproducibility.

To strengthen the manuscript, we have demonstrated EpiVerse's reproducibility in two ways. First, we replicated the training process three times and found that the performance was consistently reliable across all runs, which we have shown in Supplementary Figure 6.

Supplementary Figure 5 EpiVerse shows high reproducibility.

After three retraining runs, EpiVerse's performance remained highly consistent on four evaluation metrics.

Second, we have made the source code publicly available at <https://github.com/jhhung/EpiVerse> and provided pre-trained model weights at <https://zenodo.org/records/13759557>, allowing others to reproduce our results.

7. The authors mention that EpiVerse uses substantial GPU memory requirements for processing. It would be beneficial to actually mention the computational time and GPU memory needed to run this pipeline. As well as having a comparison table/figure that showcases the computational time and GPU memory from EpiVerse and the other methods that were being compared would strengthen the

manuscript.

We appreciate the reviewer's suggestion. We measured the whole genome inference runtime on GPU for EpiVerse, C.Origami, and Orca (1Mb-model), while excluding HiC-Reg due to its CPU-only implementation. The experimental setup involved an NVIDIA Tesla V100 GPU and 32 cores of an Intel(R) Xeon(R) Gold 6154 CPU. The comparison results, including computational time and GPU memory usage, have been added in Supplementary Table 8.

Supplementary Table 3 Whole Genome Inference time comparison between EpiVerse, C.Origami, and Orca. We compare the running times of the three models on a NVIDIA Tesla V100 GPU and 32 cores of an Intel(R) Xeon(R) Gold 6154 CPU. HiC-Reg was excluded due to its CPU-only implementation.

Methods	GPU Memory	Inference Time (s)
EpiVerse - HiConformer	7979 Mb	39213
EpiVerse - MIRNet	9005 Mb	5760
C.Origami	4223 Mb	79966
Orca (1Mb-model)	2283 Mb	1203

Reviewer #3 (Remarks on code availability):

README file is very well organized, well explained, and user friendly.

We sincerely thank the reviewer for their positive feedback. During this revision, we also updated the GitHub repository to include additional information and scripts to make EpiVerse more accessible.

Point-by-Point Response to Reviewer Comments

Reviewer #1 (Remarks to the Author):

Comment 1: Regarding the concern about using scikit-image's resize function to adjust the outputs of C.Origami and Orca to match EpiVerse's resolution, the authors explained that this approach was consistent with the original C.Origami study. However, this reasoning is not convincing. In the original C.Origami paper, the ground truth matrices were at ~8 kb resolution, while Akita predicted matrices at ~2 kb resolution and Orca at ~4 kb resolution. In that context, resizing simply coarsens the matrices to align with the lower-resolution ground truth, which is reasonable. However, in the current study, the ground truth matrices are at 5 kb resolution, and the authors resized C.Origami-predicted matrices (originally at 8 kb) to this finer resolution. A coarser resolution matrix inherently lacks the interaction details necessary for finer resolution analysis, and resizing such matrices to finer resolutions is likely to artificially degrade their performance. To ensure a fair comparison, the authors should calculate performance metrics for each method prior to resizing, especially for C.Origami. Comparisons should be made using ground truth matrices at matching resolutions (e.g., 8 kb for C.Origami and 4 kb for Orca). Alternatively, I suggest the authors include a supplementary analysis by retraining EpiVerse models at resolutions consistent with C.Origami. This would allow direct comparisons without resizing. Addressing the resolution discrepancy properly is critical to substantiate the claim that EpiVerse outperforms C.Origami.

Response: Thank you for the insightful suggestion. We acknowledge the challenges in ensuring fairness when comparing tools with different native resolutions (C.Origami at 8 kb, Orca at 4 kb, and EpiVerse at 5 kb). As the reviewer rightly pointed out, resizing C.Origami-predicted matrices to a finer resolution could introduce artifacts and potentially degrade performance. We appreciate the importance of comparing methods at their native resolutions to avoid such biases. Accordingly, we conduct the analysis using resolutions specific to each method, as suggested by the reviewer, to ensure a more accurate and fair comparison.

In brief, for the comparison between EpiVerse and C.Origami, we now extract the corresponding 1Mb region from C.Origami at its native 8 kb resolution without resizing. Since resizing the matrices of EpiVerse predictions and the ground truth from a finer resolution (i.e., 5 kb) to a coarser resolution (8 kb) is less likely to introduce artifacts, we use the scikit-image resize function to adjust them to match the 8 kb resolution. All matrices were subsequently recalculated, and the results of the comparison between EpiVerse and C.Origami are presented in the figures below.

Similarly, for the comparison between EpiVerse and Orca, we retain Orca's predictions at their native 4 kb resolution without resizing and use the scikit-image resize function to adjust other matrices to match the 4 kb resolution. The results of this comparison are also provided in the figures below.

As demonstrated in the figures above, the conclusions that EpiVerse outperforms the other tools remain consistent with our claims.

Comment 2: I appreciate the addition of active enhancers to Figure 3d and Supplementary Figure 10. However, I am surprised that, in some cell lines (e.g., K562 in Figure 3d and the leftmost columns for each region in Supplementary Figure 10), the entire region appears to be covered by active enhancers. Including H3K27ac signal tracks as a reference would enhance the interpretation of these regions.

Response: We appreciate this suggestion. To enhance the interpretability of Figure 3d and Supplementary Figure 10, we overlaid H3K27ac signal tracks as a reference. This addition will provide a clearer context for the coverage of active enhancers and help in interpreting the chromatin landscape.

The updated Figure 3 and its legend:

Figure 1 EpiVerse identifies elements defining chromatin structure. a. DNA Motif Analysis Results: The top hit motifs derived from 64 filters, showcasing the significant DNA sequences identified through the motif analysis. **b. Gene Set Enrichment Analysis (GSEA) Using IG for Chromosome 21:** Chart displaying the enrichment scores of genes based on their surrounding epigenetic signal importance as determined by IG for chromosome 21. **c. Epigenetic Signals' Tissue Specificity in Enhancer State:** Scatter plot delineating the tissue specificity of various epigenetic signals in their influence on enhancer states, with proximity to the diagonal line indicating the degree of tissue specificity. **d. Largest Differential Enhancer Regions:** The heatmaps display the chromatin interaction differences, highlighting regions with significantly varied enhancer activity between the two cell lines. Stronger chromatin looping interactions are evident in K562 as compared to GM12878. The top row represents the ground truth chromatin interactions, while the bottom row shows the imputed data. The rightmost column presents the differential heatmaps, where red indicates regions with stronger interactions in GM12878 and blue denotes stronger interactions in K562. Additionally, the figure includes active enhancer states along with H3K27ac and CTCF signal tracks for the same genomic region. Yellow bars indicate shared enhancer states between the two cell lines.

The updated Figure S10 and its legend:

“Supplementary Figure 10 Comparative Visualization of Enhancer Variation Regions. Each panel presents Hi-C contact maps for selected genomic regions, showcasing the distribution of active enhancers alongside H3K27ac signal tracks. Progressing from left to right, the panels illustrate a decline in the count of active enhancers, paralleled by a corresponding decrease in Hi-C interaction frequencies.”

Comment 3: In my previous comments, I raised concerns about Figures 5d and 5e being unrelated to chromatin interactions. In this revision, the authors explained that these panels demonstrate the ability of imputed ChromHMM data to identify tissue-specific upregulated genes. However, these analyses appear to highlight the capabilities of Avocado rather than EpiVerse, as identifying ChromHMM chromatin states is straightforward when epigenetic signals are available. A more compelling analysis would involve investigating whether regions predicted by EpiVerse to undergo dramatic 3D genomic changes are enriched for differentially expressed genes. Furthermore, Figure 5e presents a GSEA analysis of the top 200 upregulated genes marked by active enhancers in Capan-1. How do the results compare when analyzing the top 200 upregulated genes without considering active enhancers? Again, instead of focusing on differentially expressed genes, it would be more informative to determine whether dynamic changes in 3D genome structure alone can identify genes enriched in pathways related to cell motility and cancer metastasis.

Response: Thank you for raising these important points and for encouraging a more in-depth exploration of the relationship between chromatin dynamics and gene regulation. The HiConformer model within our EpiVerse framework is designed to leverage multitask learning to simultaneously predict ChromHMM states and Hi-C contact maps. This approach uses a shared embedding to project both chromatin states and 3D structures, inherently linking predicted enhancer states to chromatin dynamics. Consequently, the enhancer states predicted by EpiVerse provide structural insights that go beyond the capabilities of Avocado, as they reflect both chromatin states and interactions within a unified framework.

To address the suggestion to analyze dynamic chromatin interaction changes, we updated the analyses presented in Figures 5d and 5e. In our original Figure 5d, we tallied the number of enhancer states around differentially expressed genes and ranked regions accordingly. However, tallying chromatin interaction changes associated with differentially expressed genes presents additional challenges, as structural changes can occur both in proximity to and at a distance from the associated genes, making it difficult to define the precise range of the affected domain. To address this, for Figure 5d, we calculated chromatin interaction differences for each 5 Kbp bin within a ± 15 -bin (150 Kbp) window surrounding differentially expressed genes. Significant interaction changes, defined as exceeding thresholds of more than 3 or less than -3 normalized contact counts, were assigned binary values, and these values were summed across the window to compute a Hi-C interaction change score for each region. Regions were then ranked independently by Hi-C interaction scores and by active enhancer differences, which were determined by the differential count of bins exhibiting active enhancer states. By intersecting these rankings, we evaluated the proportion of differentially expressed genes located within regions enriched for both Hi-C and enhancer changes. Statistical significance of the overlap was confirmed through permutation testing (p-value: 0.0027).

For Figure 5e, we extended this approach to rank differentially expressed genes based solely on Hi-C interaction change scores, without considering enhancer activity. Using GSEA, we found that pathways linked to cell motility, metastasis, and apoptotic processes were enriched among genes with higher chromatin interaction changes, underscoring the biological relevance of structural chromatin dynamics. Notably, these findings strengthen the connection between EpiVerse-predicted chromatin changes and biologically significant pathways, addressing the

concerns regarding the role of dynamic chromatin interactions in identifying differentially expressed genes.

These new results demonstrate EpiVerse's ability to integrate chromatin state and structural dynamics, providing biologically meaningful insights into the interplay between 3D genome architecture and gene regulation. We believe this addresses the reviewer's concerns comprehensively and highlights the utility of EpiVerse for advancing the understanding of chromatin dynamics in cancer progression.

The detailed description of the steps associated with Figures 5d and 5e has been added to the Methods section, as outlined below:

“Assessing the link between differentially expressed genes and chromatin dynamics

Hi-C interaction differences between the two cell lines were calculated for each 5,000 bp bin by considering contact counts within a ± 15 -bin window associated with each region. For each bin, significant interaction changes, defined as exceeding thresholds of more than 3 or less than -3 normalized contact count differences, were assigned a binary value of 1, while all other changes were marked as 0. The values of all bins within the associating window were summed to compute a summary Hi-C interaction change score for each region. The highest possible scores for a region were 28, since the predicted Hi-C contact map does not consider the immediate vicinity (± 1 bin) of a region.

Regions were independently ranked based on their Hi-C interaction scores and active enhancer differences, with the latter determined by the differential count of bins exhibiting the active enhancer state. The intersection of these ranked lists was used to evaluate the number of differentially expressed genes located in regions enriched for both Hi-C and enhancer changes. To account for potential regulatory influences, gene boundaries were extended by ± 0.6 Mb during the analysis. Statistical significance of the overlap was assessed through a permutation test, in which regions were randomly shuffled 10,000 times to calculate a p-value for the observed associations.

Ranking differentially expressed genes by Hi-C interaction changes

Differentially expressed genes between Capan-1 and PANC-1 cells were mapped to genomic regions at a 1 Mb resolution. The Hi-C interaction change score of all bins within a ± 0.6 Mb range of each gene were summed and then used to rank genes. This ranked list served as input for GSEA to identify GO terms under the biological process ontology associated with the top differentially expressed genes.”

The main text related to Figures 5d and 5e has been revised:

“To investigate the relationship between chromatin interaction changes and differentially expressed genes, we analyzed structural changes in chromatin interactions near the differentially

expressed genes between Capan-1 and PANC-1 cells. Specifically, we examined the proportion of differentially expressed genes located within genomic regions enriched for significant chromatin interaction changes and enhancer activity differences. Regions with overlapping chromatin interaction changes and enhancer activity were more likely to contain differentially expressed genes, and statistical testing confirmed the significance of this observation (see Figure 5d; p -value: 0.0027 by permutation test). Differentially expressed genes were further ranked based on chromatin interaction changes, and GSEA identified GO terms associated with cell motility, metastasis, and apoptotic processes (Figure 5e).”

The updated Figure 5 and its legend:

Figure 2 In silico perturbational pancreatic cancer metastasis Hi-C experiment

a EpiVerse Pipeline for PANC Experiment Reproduction: Schematic diagram illustrating the EpiVerse process from constructing virtual epigenetic signals to chromatin 3D structure modeling, using various histone modifications and CTCF as inputs to accurately replicate chromatin structures. **b Comparison between Imputation Hi-C and experimental Hi-C:** Heatmaps indicating the Hi-C interaction matrices with substantial variations across chromosome 15. The top row presents the ground truth Hi-C matrices for Capan-1 and PANC, and their differential heatmap, while the bottom row displays imputed data. **c Enhanced Predictive Performance with Additional Tracks:**

*A bar graph comparing the Spearman correlation coefficient of EpiVerse's predictive accuracy for Capan-1 and PANC-1 datasets. The graph highlights improvements as additional epigenetic tracks are integrated: 1 track (CTCF), 2 tracks (CTCF + H3K4me3), 4 tracks (CTCF + H3K4me3 + H3K9me3 + H3K27ac), and 6 tracks (CTCF + H3K4me3 + H3K9me3 + H3K27ac + H3K27me3 + H3K36me3). **d Proportion of Differentially Expressed Genes in Regions Enriched with Chromatin Interaction and Enhancer Activity Changes:** A gene identification curve showing the proportion of differentially expressed genes located in regions enriched for both top enhancer and Hi-C signal differences in Capan-1, compared against random regions. EpiVerse consistently identifies a higher proportion of relevant genes, outperforming the random baseline. **e Enriched GO Terms of Differentially Expressed Genes Ranked by Chromatin Interaction Changes:** GSEA of the top-ranked differentially expressed genes between Capan-1 and PANC-1 cells, revealing enrichment in GO terms associated with cell motility, metastasis, and apoptotic processes.*

Comment 4: The manuscript requires careful proofreading to ensure clarity and coherence. For example, the caption for Figure 5b includes the following sentence: “Gene Discovery via Active Enhancer Counts: Line graph demonstrating the discovery rate of genes using the active enhancer count approach, validating its effectiveness in identifying genes with differential expression.” This description appears irrelevant to the figure content. Such inconsistencies should be corrected to improve the overall quality of the manuscript.

Response: Thank you for pointing out this issue. We acknowledge that the caption for Figure 5b mistakenly included an irrelevant sentence due to a copy-and-paste error. This sentence has now been removed to ensure the caption accurately reflects the figure content. We have also thoroughly proofread the manuscript to address similar inconsistencies and improve clarity and coherence throughout.

Reviewer #1 (Remarks on code availability):

Comment: The authors have done a great job updating both the code and the documentation. The software can now be installed and run successfully.

Response: Thank you for this positive feedback. We are glad that the updated code and documentation met your expectations.

Reviewer #2 (Remarks to the Author):

Comment: The author has addressed all of our questions and concerns, and we are satisfied with the response.

Response: Thank you for your supportive comments. We are pleased that our responses and revisions have satisfactorily addressed your concerns.

Reviewer #2 (Remarks on code availability):

Comment: Yes, I was able to install and run the code successfully. The provided README file contained sufficient instructions for installation and execution, making the process straightforward. The code is well-structured and serves as a valuable resource for the community.

Response: Thank you for this encouraging feedback. We are delighted that the updated README and code structure facilitated a smooth installation and execution process.

Reviewer #3 (Remarks to the Author):

Comment: I thank the authors for addressing all the questions and concerns that I had. They have responded very clearly and well to the questions, and I recommend accepting this article once they address some minor issues about their GitHub.

Response: Thank you for your positive feedback and recommendation for acceptance. Regarding the minor issue of the warning messages, we investigated and found that the warnings stem from upstream dependencies beyond our direct control (explained in the response of the next comment). While we are unable to resolve this issue at its source, we confirm that the warnings do not affect the functionality of the environment. We sincerely hope this addresses your concern.

Reviewer #3 (Remarks on code availability):

Comment: The authors have significantly improved the GitHub with thorough explanations and clear instructions. The interactive browser, EpiVerse HiCViewer, is a nice touch. For HiConformer training and MIRNet training Environment and Data processing environment, it would be nice to have lines "git clone https://github.com/jhhung/EpiVerse.git; cd EpiVerse" before the conda env create comment to lessen the confusion. The authors should also list the recommended conda version that is needed to create the environments as I'm getting a warning message (conda 23.7.4): [The warning messages from conda about version specifications have been removed from this comment for clarity.]

Response: Thank you for the detailed feedback on our GitHub. Regarding the warning messages, these indicate that some dependencies use version specifications with .* (e.g., 1.9.0.*), which are now deprecated. Conda automatically treats these versions as 1.9.0, ignoring the .*. Upon investigation, we found that these dependencies are indirectly required through upstream packages and are not directly specified in the HiConformer environment file. Unfortunately, as these dependencies are beyond our control, we are unable to resolve the warnings. However, we confirm that they do not affect the functionality of the environment.

We would like to express our sincere gratitude to the reviewers and the editor for their dedicated time and constructive feedback.

=====

REVIWER COMMENTS

Reviewer #1 (Remarks to the Author):

The authors have thoroughly addressed all of my previous concerns. I have no further concerns and fully support the acceptance of this manuscript in its current form.

Reviewer #1 (Remarks on code availability):

The software can be installed and run successfully.

Response:

We sincerely appreciate Reviewer #1's positive feedback and support for the acceptance of our manuscript. We are also grateful for the confirmation that the software can be installed and run successfully. Thank you for your time and thoughtful review.